# Generalizing Supervised Contrastive Learning: A Projection Perspective

## Abstract

Self-supervised contrastive learning (SSCL) has emerged as a powerful paradigm for representation learning and has been studied from multiple perspectives, including mutual information and geometric viewpoints. However, supervised contrastive (SupCon) approaches have received comparatively little attention in this context: for instance, while InfoNCE used in SSCL is known to form a lower bound on mutual information (MI), the relationship between SupCon and MI remains unexplored. To address this gap, we introduce ProjNCE, a generalization of the InfoNCE loss that unifies supervised and self-supervised contrastive objectives by incorporating projection functions and an adjustment term for negative pairs. We prove that ProjNCE constitutes a valid MI bound and affords greater flexibility in selecting projection strategies for class embeddings. Building on this flexibility, we further explore the centroid-based class embeddings in SupCon by exploring a variety of projection methods. Extensive experiments on image and audio datasets demonstrate that ProjNCE consistently outperforms both SupCon and standard cross-entropy training. Our work thus refines SupCon along two complementary perspectives—information-theoretic and projection viewpoints—and offers broadly applicable improvements whenever SupCon serves as the foundational contrastive objective.

## 1 Introduction

Contrastive learning (Chen et al., 2020; Tian et al., 2020) has recently emerged as a highly effective paradigm for representation learning, achieving strong performance across diverse domains such as computer vision (Wang et al., 2021), bioinformatics (Yu et al., 2023), and multimodal processing (Radford et al., 2021; Girdhar et al., 2023). In the self-supervised setting, contrastive losses (Oord et al., 2018; Chen et al., 2020; Chuang et al., 2020; Barbano et al., 2022; Chopra et al., 2005; Hermans et al., 2017) treat each data instance as its own "class" and aim to pull together multiple augmentations of the same instance while pushing apart different instances, attempting to learn representations that capture meaningful invariances. Supervised contrastive (SupCon) learning (Khosla et al., 2020) extends this idea to labeled data by grouping examples of the same class as positive pairs, often surpassing the performance of standard cross-entropy (CE) in downstream classification tasks.

Despite these successes, SupCon has not been clearly related to mutual information, whereas the InfoNCE is known to provide a lower bound on mutual information (Poole et al., 2019), implying that it maximizes a mutual information lower bound through minimizing the InfoNCE. This discrepancy raises a natural question: *How is the SupCon loss relevant to the mutual information $I(\mathbf{X}; C)$ between input features and class labels?* This question is fundamental in understanding the SupCon.

Moreover, from the alignment and uniformity perspective (Wang & Isola, 2020), SupCon forces alignment using the positive pair as a centroid of embeddings of the same class. Similarly, a natural question arises: *Is the centroid-based class embedding the best we can select?* Since the class embedding is a basis of every SupCon-based methods, potential refinement of it benefits broad SupCon-based methods and applications.

In this work, to relate SupCon to mutual information, we propose ProjNCE, a generalization of InfoNCE based on a projection perspective. By introducing an additional adjustment term for negative pairs, ProjNCE becomes a valid lower bound on mutual information. We show that SupCon can be

(approximately) viewed as a special case of ProjNCE, though SupCon itself does not necessarily preserve the mutual-information bound.

Regarding the question of ideal class embedding, we explore projection functions in ProjNCE to replace rigid centroid-based class embeddings. In particular, we study orthogonal projection that is optimal in $\ell_2$ projection error, median based projection that is robust to outlier, and MLP-based projection that does not require hyper-parameter tuning. For the orthogonal projection, we approximate it using soft labels, i.e., posterior class probabilities given the features. When such soft labels are not available, we estimate them via a kernel-based estimator, specifically the Nadaraya-Watson estimator (Nadaraya, 1964; Watson, 1964). We validate our proposed methods through extensive experiments on vision and audio datasets. The results demonstrate that ProjNCE with various projections outperform baselines in most settings.

In summary, our contributions are threefold:

1. We *generalize contrastive loss* to unify supervised and self-supervised contrastive learning under projection perspective. The generalized loss, namely ProjNCE, is flexible to choose arbitrary projection function.

2. We theoretically analyze SupCon and ProjNCE under information-theoretic viewpoint and *show that minimizing ProjNCE can maximize mutual information.*

3. We explore standout projection functions of ProjNCE and observe performance gain, *opening a new avenue for enhancing contrastive objectives.*

4. We conduct comprehensive experiments on multiple vision and audio datasets, showing that *ProjNCE with proper projection function consistently outperforms baselines*, positioning ProjNCE as a promising foundation for future supervised contrastive learning approaches.

## 2 SUPERVISED CONTRASTIVE LEARNING

In this section, we first introduce our notation and review the SupCon and InfoNCE losses. We then pose two key questions—from an information-theoretic perspective and from the lens of alignment and uniformity. Building on these viewpoints, we propose a variant of the InfoNCE loss that generalizes SupCon, thereby linking SupCon to mutual information. Finally, we analyze SupCon under this unified loss framework.

### 2.1 PRELIMINARIES AND MAIN QUESTIONS

For an $M$-class classification problem, let $(\boldsymbol{x}, \boldsymbol{c}) \sim p(\boldsymbol{x}, \boldsymbol{c})$ be an input feature and the corresponding label pair. We denote by $\mathcal{B} = \{(\boldsymbol{x}_i, \boldsymbol{c}_i)\}_{i=1}^{N}$ a mini-batch of size $N$ containing randomly sampled pairs of feature and label from a dataset. SupCon loss (Khosla et al., 2020) is defined as

$$I_{\text{NCE}}^{\text{sup}} = -\mathbb{E}\left[\frac{1}{|\mathcal{P}(i)|} \sum_{p \in \mathcal{P}(i)} \log \frac{\exp(\boldsymbol{z}_i \cdot \boldsymbol{z}_p/\tau)}{\sum_{j \in \mathcal{B} \setminus \{i\}} \exp(\boldsymbol{z}_i \cdot \boldsymbol{z}_j/\tau)}\right], \tag{1}$$

where the expectation is taken with respect to $\prod_{j=1}^{N} p(\boldsymbol{x}_j, \boldsymbol{c}_j)$, $\boldsymbol{z}_i = f(\boldsymbol{x}_i)$ is the embedding of $\boldsymbol{x}_i$ by an encoder $f : \mathcal{X} \to \mathcal{S}^{d_z - 1}$, $\boldsymbol{z}_i \cdot \boldsymbol{z}_j$ denotes the inner (dot) product, and $\mathcal{P}(i) = \{k \in [N] \setminus \{i\} : \boldsymbol{c}_k = \boldsymbol{c}_i, \boldsymbol{x}_k \in \mathcal{B}\}$ is the set of indices of features associated with class $\boldsymbol{c}_i$ in the mini-batch except $i$.

The SupCon loss (1) is a straightforward modification of the InfoNCE loss (Oord et al., 2018) for supervised learning. While the InfoNCE loss is a well-established estimator of mutual information (Poole et al., 2019), the relationship between SupCon loss and mutual information has not yet been explicitly established. This raises the natural question (Q1) concerning the relation between the SupCon and preserving the class-information contained in the features:

Q1: *How is the SupCon loss relevant to the mutual information $I(\mathbf{X}; C)$ between input features and class labels?*

In Section 2.2 we address this question by analyzing the SupCon loss from an information-theoretic perspective and derive a variant of the SupCon loss inspired by the multi-sample lower bound of mutual information (Nguyen et al., 2010; Poole et al., 2019).

Complementing to the mutual information perspective, the concepts of alignment and uniformity offer valuable interpretations of contrastive loss (Wang & Isola, 2020). The $i$-th SupCon loss is:

$$I_{\text{NCE},i}^{\text{sup}} = -\frac{z_i}{\tau} \cdot \sum_{p \in \mathcal{P}(i)} \frac{z_p}{|\mathcal{P}(i)|} + \log \sum_{j \in \mathcal{B}\setminus\{i\}} e^{z_i \cdot z_j/\tau}, \tag{2}$$

where the first term corresponds to the alignment, and the second term expresses the uniformity. Compared with the $i$-th InfoNCE loss in self-supervised learning,

$$I_{\text{NCE},i}^{\text{self}} = -\frac{z_i \cdot z_p}{\tau} + \log \sum_{j \in \mathcal{B}\setminus\{i\}} e^{z_i \cdot z_j/\tau}, \tag{3}$$

the difference in the SupCon loss arises from the alignment term, which uses the average embedding of $z_p$, $p \in \mathcal{P}(i)$—referred to as the centroid of $c_i$-class embeddings—instead of a single embedding $z_p$.

Intuitively, aligning $z_i$ with the centroid helps cluster embeddings within the same class when the labels are reliable (e.g., noiseless) and the features $x_p, p \in \mathcal{P}(i)$ are equally informative. However, when the features $x_p, p \in \mathcal{P}(i)$ are not equally informative for the class or labels are not reliable, aligning the embedding with the centroid becomes problematic. For instance, in the case of noisy labels, if a majority of the labels are randomly permuted across class categories, $\mathcal{P}(i)$ becomes unreliable, and the centroid may fail to capture the correct class representation. Furthermore, if the dependency between $x_p$ and $c_p$ varies across $p \in \mathcal{P}(i)$, the uniform weighting used in the centroid fails to account for these disparities in information.

Motivated by these observations, our second question is:

Q2: *Is the centroid-based class embedding the best we can select?*

In Section 3, we investigate various approaches for class embedding through the lens of projection.

## 2.2 PROJNCE: A VALID MUTUAL INFORMATION LOWER BOUND

We relate SupCon loss to the mutual information between the embedding and the class, denoted $I(\mathbf{Z}; C)$. We begin with the InfoNCE loss (Oord et al., 2018):

$$I_{\text{NCE}}^{\text{self}}(\mathbf{Z}; C) = \frac{1}{N} \sum_{i=1}^{N} \mathbb{E}_P \left[ -\log \frac{e^{\psi(f(x_i), c_i)}}{\sum_{j=1}^{N} e^{\psi(f(x_i), c_j)}} \right], \tag{4}$$

where $P = p(x_i|c_i) \prod_{j=1}^{N} p(c_j)$. Here, $\psi(x, y)$ is a critic that measures similarity between $x$ and $y$.

A direct similarity comparison between the input feature $x \in \mathcal{X}$ and the class $c \in [1 : M]$ is infeasible unless both are projected to the same space. Typically, input pairs $(x, c)$ are projected to $\mathbb{R}^{d_z}$ using encoders, followed by normalization and the evaluation of cosine similarity.

To generalize this approach and relate it to $I(\mathbf{Z}; C)$, we define projection functions $g_+(c)$ and $g_-(c)$ that map $c$ into $\mathbb{R}^{d_z}$. Substituting these projection functions into (4) results in:

$$I_{\text{NCE}}^{\text{self-p}}(\mathbf{Z}; C) = \frac{1}{N} \sum_{i=1}^{N} \mathbb{E}_P \left[ -\log \frac{e^{\psi(f(x_i), g_+(c_i))}}{\sum_{j=1}^{N} e^{\psi(f(x_i), g_-(c_j))}} \right]. \tag{5}$$

Unlike (4), the projection-based InfoNCE in (5) allows for different projections for positive ($g_+(c)$) and negative ($g_-(c)$) samples, leading to the following properties:

- (5) generalizes both SupCon loss in supervised settings and InfoNCE loss in self-supervised settings. For instance: 1) setting $g_+(c_i) = \sum_{p \in \mathcal{P}(i)} \frac{f(x_p)}{|\mathcal{P}(i)|}$ and $g_-(c_j) = f(z_j)$, $\forall j \neq i$, recovers the SupCon loss; 2) setting $g_+ = g_-$ retrieves the InfoNCE loss.
- The choice of $g_+$ and $g_-$ introduces inductive biases, allowing flexibility in designing contrastive learning objectives tailored to specific tasks.

Poole et al. (2019) proved that the InfoNCE loss provides a lower bound on mutual information by leveraging a multi-sample variant of the standard lower bound (Nguyen et al., 2010). In a similar manner, we derive lower bounds on $I(\mathbf{X}; C)$ for the projection-based InfoNCE loss defined in (5) and for the SupCon loss:

**Proposition 2.1.** *For any $g_+$ and $g_-$, the projection incorporated InfoNCE in* (5) *bounds mutual information as*

$$I(\mathbf{X}; C) \geq 1 + \log N - I_{\mathrm{NCE}}^{\mathrm{self\text{-}p}}(\mathbf{X}; C) - R(\mathbf{X}, C), \tag{6}$$

*where*

$$R(\mathbf{X}, C) = \mathbb{E}_{p(\boldsymbol{x}) \prod_{j=1}^{N} p(\boldsymbol{c}_j)} \left[ \frac{\sum_{k=1}^{N} e^{\psi(f(\boldsymbol{x}), g_+(\boldsymbol{c}_k))}}{\sum_{k=1}^{N} e^{\psi(f(\boldsymbol{x}), g_-(\boldsymbol{c}_k))}} \right]. \tag{7}$$

*Proof.* The proof is in Appendix B.1 $\qquad\square$

**Corollary 2.2.** *SupCon loss bounds mutual information as*

$$I(\mathbf{X}; C) \geq 1 + \log N - I_{\mathrm{NCE}}^{\mathrm{sup}}(\mathbf{X}; C) - R^{\mathrm{sup}}(\mathbf{X}, C), \tag{8}$$

*where* $R^{\mathrm{sup}}(\mathbf{X}, C) = \mathbb{E}_{p(\boldsymbol{x}) \prod_{j=1}^{N} p(\boldsymbol{c}_j)} \left[ \frac{\sum_{k=1}^{N} e^{\psi(f(\boldsymbol{x}), \sum_{p \in \mathcal{P}(k)} \frac{f(\boldsymbol{x}_p)}{|\mathcal{P}(k)|})}}{\sum_{k=1}^{N} e^{\psi(f(\boldsymbol{x}), f(\boldsymbol{x}_k))}} \right].$

*Proof.* Setting $g_+(\boldsymbol{c}_k) = \sum_{p \in \mathcal{P}(k)} \frac{f(\boldsymbol{x}_p)}{|\mathcal{P}(k)|}$ and $g_-(\boldsymbol{c}_k) = f(\boldsymbol{x}_k)$ in $I_{\mathrm{NCE}}^{\mathrm{self\text{-}p}}(\mathbf{X}; C)$ and $R(\mathbf{X}, C)$ in Proposition 2.1 gives the bound, which completes the proof. $\qquad\square$

Proposition 2.1 establishes the connection between (5) and mutual information through the derived lower bound. Due to the presence of the adjustment term $R(\mathbf{X}, C)$, the SupCon loss, when derived by selecting the corresponding projection functions in (5) (as demonstrated in Corollary 2.2), might not serve as a strict lower bound for mutual information. Consequently, minimizing the SupCon loss does not necessarily guarantee an increase in mutual information.

To address this limitation, we incorporate the adjustment term $R(\mathbf{X}, C)$ into (5), thereby ensuring that it becomes a proper lower bound, as validated by Proposition 2.1. This leads us to propose a novel variant of the InfoNCE loss, referred to as the ProjNCE loss:

**Definition 2.3** (ProjNCE). The ProjNCE loss is defined as

$$I_{\mathrm{proj}}(\mathbf{Z}; C) = I_{\mathrm{NCE}}^{\mathrm{self\text{-}p}}(\mathbf{Z}; C) + R(\mathbf{Z}, C), \tag{9}$$

where $I_{\mathrm{NCE}}^{\mathrm{self\text{-}p}}(\mathbf{Z}; C)$ is in (5), and $R(\mathbf{X}, C)$ is in (7).

Since the negative of the ProjNCE loss provides a valid lower bound for mutual information (Proposition 2.1), mutual information can be effectively maximized by minimizing the ProjNCE loss.

While $I_{\mathrm{NCE}}^{\mathrm{self\text{-}p}}(\mathbf{Z}; C)$ can be computed analogously to the standard InfoNCE loss, evaluating $R(\mathbf{X}, C)$ requires the mutual independence of $\boldsymbol{x}$ and $\boldsymbol{c}_j$, $j \in [N]$. In practice, we approximate $R(\mathbf{X}, C)$ via a leave-one-out scheme: for each sample, we exclude its true class and use only independently drawn $\boldsymbol{c}_j$. Throughout this paper, we estimate $R(\mathbf{X}, C)$ from a mini-batch using the following approximation:

$$R(\mathbf{X}, C) \approx \frac{1}{N} \sum_{i=1}^{N} \mathbb{E}_P \left[ \frac{\sum_{k \neq i} e^{\psi(f(\boldsymbol{x}_i), g_+(\boldsymbol{c}_k))}}{\sum_{k \neq i} e^{\psi(f(\boldsymbol{x}_i), g_-(\boldsymbol{c}_k))}} \right], \tag{10}$$

where $P$ is the same distribution used in (4) and (5). In what follows, although ProjNCE (9) can employ arbitrary projection functions $g_+$ and $g_-$, we will use the term "ProjNCE" to denote the special case in which these projections coincide with those used in SupCon.

## 2.3 THE ADJUSTMENT TERM $R(\mathbf{X}; C)$

From the perspective of alignment and uniformity (Wang & Isola, 2020), we analyze the $i$-th term of the ProjNCE loss in Definition 2.3, which can be expressed as:

$$I_{\mathrm{proj,i}} = \mathbb{E}_P \left[ -\psi(f(\boldsymbol{x}_i), g_+(\boldsymbol{c}_i)) + \log \sum_{j=1}^{N} e^{\psi(f(\boldsymbol{x}_i), g_-(\boldsymbol{c}_j))} \right] + \mathbb{E}_Q \left[ \frac{\sum_{k=1}^{N} e^{\psi(f(\boldsymbol{x}), g_+(\boldsymbol{c}_k))}}{\sum_{k=1}^{N} e^{\psi(f(\boldsymbol{x}), g_-(\boldsymbol{c}_k))}} \right]. \tag{11}$$

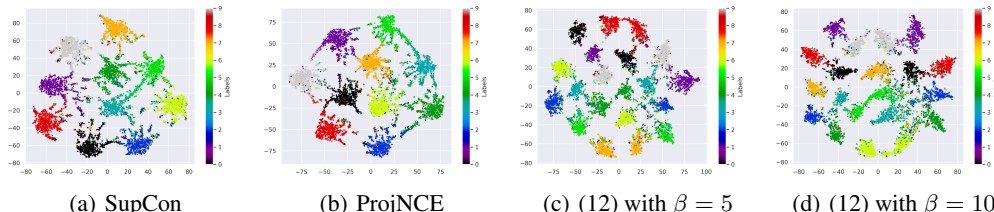

|  (a) SupCon  |  (b) ProjNCE  |  (c) (12) with $\beta = 5$  |  (d) (12) with $\beta = 10$  |

Figure 1: t-SNE plots of CIFAR-10 embeddings from Resnet-18 learned with label noise of probability 0.3. The four figures are obtained from different loss selection: (a) SupCon (b) ProjNCE (c) (12) with $\beta = 5$ (d) (12) with $\beta = 10$. The adjustment term in (7) forces the embedding clusters to spread out.

where $P = p(\boldsymbol{x}_i|\boldsymbol{c}_i) \prod_{j=1}^{N} p(\boldsymbol{c}_j)$ and $Q = p(\boldsymbol{x}) \prod_{j=1}^{N} p(\boldsymbol{c}_j)$. Throughout the paper, we occasionally denote the $P$ and $Q$ distributions for brevity and clarity in notation.

In addition to the alignment term $\mathbb{E}_P[-\psi(f(\boldsymbol{x}_i), g_+(\boldsymbol{c}_i))]$ and the uniformity term $\mathbb{E}_P[\log \sum_{j=1}^{N} e^{\psi(f(\boldsymbol{x}_i), g_-(\boldsymbol{c}_j))}]$ of InfoNCE, (11) introduces a third term, which we call adjustment term, defined as $R(\mathbf{X}, C) = \mathbb{E}_Q \left[ \frac{\sum_{k=1}^{N} e^{\psi(f(\boldsymbol{x}), g_+(\boldsymbol{c}_k))}}{\sum_{k=1}^{N} e^{\psi(f(\boldsymbol{x}), g_-(\boldsymbol{c}_k))}} \right]$. This term represents the average ratio of the similarity between $g_+$ and $g_-$. Since the distribution $Q$ independently samples $\boldsymbol{x}$ and $\boldsymbol{c}_j$, the adjustment term measures the similarity ratio for negative pairs (i.e., pairs without shared semantics).

Intuitively, minimizing the adjustment term encourages $g_-(\boldsymbol{c}_k)$ closer to $f(\boldsymbol{x})$ than $g_+(\boldsymbol{c}_k)$ for negative pairs. This reduces the perturbation caused by false positive samples, as $g_+$ becomes more robust by tending to produce smaller similarity scores for such negative pairs.

## 2.4 ANALYSIS OF $R(\mathbf{X}; C)$

We now set $g_+(\boldsymbol{c}_i) = \frac{1}{|\mathcal{P}(i)|} \sum_{p \in \mathcal{P}(i)} f(\boldsymbol{x}_p)$ and $g_-(\boldsymbol{c}_i) = \boldsymbol{x}_i$ to analyze $R(\mathbf{X}; C)$. As demonstrated earlier, incorporating the adjustment term $R(\mathbf{X}, C)$ into SupCon establishes a valid lower bound for mutual information, a linkage that can be exploited to enhance classification performance. To validate our analysis and see the effect of $R(\mathbf{X}; C)$, we perform classification tasks and compare the visualization of the embeddings learned by SupCon and ProjNCE.

Table 1 reports the classification accuracies using SupCon and ProjNCE losses. Here, ProjNCE is SupCon with the adjustment term $R$ in (9), showing the effect of $R$. We adopt ResNet-18 (He et al., 2016) as the encoder. In every epoch during training, we evaluate zero-shot classification accuracy on a held-out test set with clean labels[1]. We report the best accuracy during training encoders. Although the gains in Table 1 are modest, ProjNCE consistently outperforms SupCon—presumably because it optimizes a tighter mutual information bound. In Section 4 we present

Table 1: Top-1 classification accuracies of SupCon and ProjNCE (i.e., SupCon with the adjustment $R$ in (9)).

| DATASET | SUPCON | PROJNCE |
|---|---|---|
| CIFAR-10 | 93.86 | **94.20** |
| CIFAR-100 | 71.51 | **71.84** |
| TINY-IMAGENET | 58.50 | **58.98** |

additional experiments measuring the mutual information between learned embeddings and class labels, confirming that ProjNCE yields higher mutual information than SupCon.

Figure 1 visualizes the t-SNE (Van der Maaten & Hinton, 2008) embeddings of the CIFAR-10 test dataset using ResNet-18 trained with different loss functions under a label noise probability of 0.3. In Figures 1(a) and 1(b), the embeddings learned by SupCon and ProjNCE are displayed, showing similar clustering patterns with respect to class. To observe how the adjustment term in the ProjNCE loss influences embedding clusters, we emphasize the adjustment term by weighting it with $\beta$ of 5 and 10, with the resulting embeddings shown in Figures 1(c) and 1(d), respectively. Specifically, we use the following loss with $\beta \in \{5, 10\}$:

$$\mathcal{L} = I_{\text{NCE}}^{\text{self-p}}(\mathbf{Z}; C) + \beta R(\mathbf{Z}, C). \tag{12}$$

---

[1]We compute class centroids in the embedding space and classify new samples by their similarity to those centroids.

Compared to SupCon, embeddings trained with a weight of 5 (Figure 1(c)) exhibit increased dispersion with two clusters for each class, and those trained with a weight of 10 (Figure 1(d)) form more distinct subclusters. We attribute this behavior to improved separation of false-positive samples, which prevents the embeddings from collapsing too tightly around their centroids. We note that setting higher $\beta$ in Figure 1 is to highlight the effect of the adjustment term $R(\mathbf{X}; C)$, not to claim a better performance. Table 5 provides the corresponding accuracies for (12). Analogous to our analysis in Corollary 2.2, setting $\beta = 1$ (i.e., ProjNCE) achieves the highest accuracy among $\beta \in \{0, 1, 5, 10\}$.

## 3 PROJECTION FUNCTIONS IN PROJNCE

In this section, we investigate the projection functions $g_+$ and $g_-$ to further optimize the ProjNCE loss. Specifically, we consider: 1) An orthogonal projection of each class label into the embedding space.; 2) the median of the embeddings corresponding to the same class; and 3) MLP-based projection into the embedding space.

### 3.1 ORTHOGONAL PROJECTION

Intuitively, the projection functions $g_+$ and $g_-$ should yield a semantical representation of $c_i$ in the embedding space. From a projection perspective, a natural choice is the orthogonal projection of the class label into that space. According to the orthogonal principle (Kay, 1993), the conditional expectation

$$\bar{f}(\boldsymbol{c}) := \mathbb{E}[f(\mathbf{X})|C = \boldsymbol{c}] \in \arg\min_{g \in \mathcal{F}} \mathbb{E}\left[\|f(\mathbf{X}) - g(\boldsymbol{c})\|^2\right], \tag{13}$$

where $\mathcal{F}$ is the set of measurable functions. Hence, $\bar{f}(\boldsymbol{c})$ serves as the natural centroid of the embeddings for $\boldsymbol{c}$ (Banerjee et al., 2005). Using this, we define variants of ProjNCE and SupCon, namely ProjNCE-perp, in Definition 3.1.

**Definition 3.1** (ProjNCE-perp). ProjNCE-perp loss is defined as

$$I_{\text{proj}}^{\text{perp}} = \frac{1}{N} \sum_{i=1}^{N} \mathbb{E}_P \left[ -\log \frac{e^{\psi(f(\boldsymbol{x}_i), \overline{f}(\boldsymbol{c}_i))}}{\sum_{j=1}^{N} e^{\psi(f(\boldsymbol{x}_i), \overline{f}(\boldsymbol{c}_j))}} \right], \tag{14}$$

where $\overline{f}(\boldsymbol{c}) = \mathbb{E}[f(\mathbf{X})|C = \boldsymbol{c}]$.

Note that ProjNCE-perp (14) does not include the adjustment term since $g_+ = g_-$ results in $R(\mathbf{X}, C) = 1$. The following proposition outlines the basic properties of ProjNCE-perp.

**Proposition 3.2.** *ProjNCE-perp loss* (14) *satisfies the following:*

- *Optimal critic $\psi^\star$ that maximizes ProjNCE-perp satisfies $\psi^\star(f(\boldsymbol{x}), \overline{f}(\boldsymbol{c})) \propto \log \frac{p(\boldsymbol{c}|\boldsymbol{x})}{p(\boldsymbol{c})} + \alpha(\boldsymbol{x})$, where $\alpha(\boldsymbol{x})$ is an arbitrary function.*

- $I(\mathbf{X}; C) \geq -I_{\text{proj}}^{\text{perp}}(\mathbf{X}; C) + \log N.$

- *With an optimal critic, it holds that $\log N - I_{\text{proj}}^{\text{perp}}(\mathbf{X}; C) \overset{a.s.}{\to} I(\mathbf{X}; C)$ as $N \to \infty$.*

*Proof.* The proof is in Appendix B.2 □

**Estimator of $\overline{f}$.** Although $\overline{f}$ is the optimal projection in terms of $\ell_2$ projection error, its exact computation requires knowledge of the conditional distribution, which is generally unavailable. To overcome this, we approximate $\overline{f}$ using a kernel regression (Tsybakov, 2008). Specifically, with $\boldsymbol{z} = f(\boldsymbol{x})$, Bayes rule yields:

$$\overline{f}(\boldsymbol{c}) = \int \boldsymbol{z} \frac{p(\boldsymbol{c}|\boldsymbol{z})p(\boldsymbol{z})}{p(\boldsymbol{c})} \mathrm{d}\boldsymbol{z} = \frac{\mathbb{E}_{p(\boldsymbol{z})}\left[p(\boldsymbol{c}|\boldsymbol{z})\boldsymbol{z}\right]}{\mathbb{E}_{p(\boldsymbol{z})}\left[p(\boldsymbol{c}|\boldsymbol{z})\right]}. \tag{15}$$

Assuming that $p(\boldsymbol{c}|\boldsymbol{z})$ is accessible, $\overline{f}$ can be estimated via Monte Carlo approximations of both the numerator and the denominator in (15). The class posterior probability $p(\boldsymbol{c}|\boldsymbol{z})$ is often referred to as

the soft label for class $c$ (Hu et al., 2016; Yang et al., 2023; Cui et al., 2023b; Jeong et al., 2023; 2024). Soft labels can be obtained through knowledge distillation (Gou et al., 2021), crowdsourced label aggregation (Ishida et al., 2022; Battleday et al., 2020; Collins et al., 2022), or kernel methods (Tsybakov, 2008; Jeong et al., 2024). Leveraging soft labels has been shown to enhance robustness and improve downstream-task performance (Battleday et al., 2020).

In this work, we employ the Nadaraya-Watson (NW) estimator (Nadaraya, 1964; Watson, 1964) to approximate $p(c|z) = \mathbb{E}[\mathbb{1}\{C = c\}|z]$, based on a given dataset $\mathcal{D} = \{(x_i, c_i)\}_{i=1}^{|\mathcal{D}|}$. This estimator serves as an intermediate step in computing $\overline{f}$ in (15).[2] The NW estimator for $p(c|z)$ is defined as:

$$\mathsf{NW}_h(c; z, \mathcal{D}) = \frac{\sum_{j=1}^{|\mathcal{D}|} K_h(d(z, z_j)) \mathbb{1}\{c_j = c\}}{\sum_{j=1}^{|\mathcal{D}|} K_h(d(z, z_j))}, \tag{16}$$

where $d : \mathbb{R}^{d_z} \times \mathbb{R}^{d_z} \to \mathbb{R}_+$ is a metric, $K_h(t) = \frac{1}{h} K(\frac{t}{h})$ is a kernel $K$ with a bandwidth $h > 0$ such that the kernel $K$ has support $[0, 1]$, is strictly decreasing, is Lipschitz continuous, and $\exists\theta$, $\forall t \in [0, 1]$, $-K'(t) > \theta > 0$. Using the NW estimator in (16), $\overline{f}(c)$ can be estimated as:

$$\hat{f}(c) = \frac{\sum_{j=1}^{N} \mathsf{NW}_h(c; f(x_j), \mathcal{D}) f(x_j)}{\sum_{j=1}^{N} \mathsf{NW}_h(c; f(x_j), \mathcal{D})}. \tag{17}$$

If soft labels $p(c|z)$ are readily available, $\mathsf{NW}_h(c; f(x_j), \mathcal{D})$ in (17) can be replaced with $p(c|z_j)$.

The following proposition formally states the consistency of $\hat{f}$ in (17), demonstrating that $\hat{f}$ with the NW estimator converges to $\overline{f}$ under some assumptions.

**Proposition 3.3.** *Assume Hölder condition: there exist $C < \infty$ and $\beta > 0$, such that for all $(u, v) \in \mathcal{S}^{d_z-1}$, $|\mathsf{NW}_h(c|u; \mathcal{D}) - \mathsf{NW}_h(c|v; \mathcal{D})| \leq Cd(u, v)$, where $d$ is the metric used in the kernel. Assume that there exists $\kappa > 0$ such that $\inf_{z \in \mathcal{S}^{d_z-1}} p_z(z) \geq \kappa$, where $p_z = \frac{dF_z}{d\mu}$ is the Radon-Nikodym derivative of the distribution function $F_z$ with respect to the Lebesgue measure $\mu$ on $\mathcal{S}^{d_z-1}$. Then, as $N \to \infty$ and $h \to 0$ with $\sqrt{\frac{\ln N}{h^{d_z} N}} \to 0$, it follows that*

$$\hat{f}(c) \to \overline{f}(c). \tag{18}$$

*Proof.* The proof is in Appendix B.3 □

As indicated in Proposition 3.3, it is recommended to use a larger batch size, which not only enhances the estimation of $\hat{f}$ but also helps tighten the mutual information bound, along with selecting a lower value for the bandwidth.

## 3.2 OTHER PROJECTIONS

While the orthogonal projection $\overline{f}$ minimizes the $\ell_2$ projection error, it is sensitive to outliers. We therefore explore a median-based projection: $f_{\mathrm{med}}(c_i) = \mathrm{median}(\{f(x_p)\}_{p \in \mathcal{P}(i)})$. Here, $f_{\mathrm{med}}(c_i)$ denotes the median of all embeddings whose class equals $c_i$. In analogy to ProjNCE-perp, we call the median-based ProjNCE variant "ProjNCE-med" if ProjNCE employs median projection for both $g_+$ and $g_-$.

Table 2: Top-1 classification accuracies of SupCon and ProjNCE-perp on ImageNet.

| METHOD | IMAGENET |
|---|---|
| SUPCON | 42.89 |
| PROJNCE-MLP | **57.66** |

Furthermore, ProjNCE-perp contains a few hyper-parameters, e.g., kernel parameters, which often burden to optimize in practice. To relax the optimization burden, we also consider MLP-based ProjNCE loss, in which ProjNCE adopts MLP for both $g_+$ and $g_-$. We denote by "ProjNCE-MLP" if ProjNCE utilizes MLP-based projection.[3] In this paper, we utilize a linear embedding matrix for the projection

---

[2]While the NW estimator is widely used, alternative kernel-based methods might be more suitable under specific conditions, which we leave as a topic for future exploration.

[3]Formal definition appears in Appendix C.

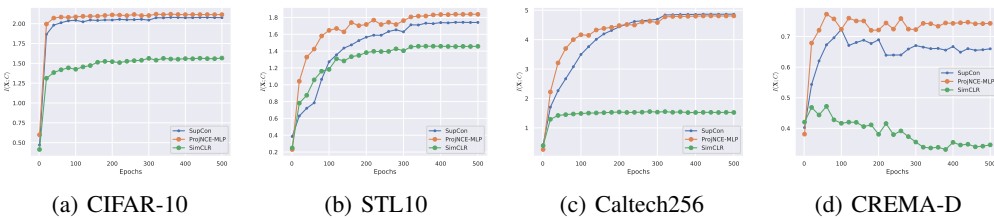

| (a) CIFAR-10 | (b) STL10 | (c) Caltech256 | (d) CREMA-D |

Figure 2: We estimate the mutual information $I(f(\mathbf{X}); C)$ between the learned embedding and the class label. ProjNCE attains higher estimated mutual information than SupCon (a tie on Caltech256), largely due to its use of a valid mutual-information bound.

in ProjNCE-MLP. Assuming the linear embedding matrix learns best possible projection, ProjNCE-MLP approximately employ the best linear projection for the class embedding. Table 2 summarizes ImageNet (Deng et al., 2009) accuracy for SupCon and ProjNCE-MLP. ProjNCE-MLP substantially outperforms SupCon, underscoring the importance of optimizing the projection function.

## 4 EXPERIMENTS

In this section, we empirically evaluate the proposed loss functions relative to cross-entropy (CE) and SupCon-based methods. First, we examine the effect of different kernel functions and bandwidth parameters in NW estimator used in ProjNCE-perp. Then, we compare the mutual information between the learned embeddings and class labels under each method. Lastly, we assess classification accuracy across various datasets, including vision and audio modalities, under feature and label noise.

### 4.1 KERNEL AND BANDWIDTH

Proposition 3.3 indicates that smaller bandwidth values generally yield more accurate estimates of $\hat{f}$ when batch sizes are large. To investigate this, we evaluate several bandwidth settings on the CIFAR-10 dataset. We employ the kernel $K(t) = 1 - t^2$ a scaled Epanechnikov variant (Epanechnikov, 1969) often favored in one-dimensional density estimation for minimizing the mean integrated squared error under certain conditions (Tsybakov, 2008). For the distance metric $d$, we compare the $\ell_1$ and $\ell_2$ distances, as well as the cosine dissimilarity $\frac{1}{2} - \frac{1}{2} \frac{\boldsymbol{u} \cdot \boldsymbol{v}}{\|\boldsymbol{u}\|\|\boldsymbol{v}\|}$.

Figure 3 in Appendix D.1 plots CIFAR-10 and STL10 accuracy versus bandwidth $h$ for ProjNCE-perp using $\ell_1, \ell_2$, and cosine dissimilarities. Accuracy is broadly similar across metrics—consistent with Proposition 3.3—with a single anomaly at $(h = 0.7, d = \cos)$ on STL10. This pattern indicates that ProjNCE-perp needs minimal kernel-parameter tuning; testing a few values typically suffices. Accordingly, we perform minimal bandwidth tuning by restricting $h \in \{0.1, 0.3, 0.5, 0.7, 0.9\}$ and using the $\ell_1$ distance for all subsequent experiments.

### 4.2 MAXIMIZING MUTUAL INFORMATION

In Figure 2, we compare the mutual information between the learned embeddings $f(\mathbf{X})$ and class labels $C$ for SupCon, ProjNCE-MLP, and SimCLR. Because the true data distribution is unknown, we estimate $I(f(\mathbf{X}); C)$ using the Mixed KSG estimator (Gao et al., 2017), which is appropriate for continuous–discrete pairs. Across datasets, ProjNCE-MLP attains the highest estimated mutual information (with a near tie with SupCon on Caltech256), while SimCLR consistently lags behind. These results align with our information-theoretic analysis: unlike SupCon, ProjNCE optimizes a valid lower bound on mutual information (Proposition 3.2), yielding embeddings that retain more class information.

### 4.3 CLASSIFICATION ACCURACY

Table 3 reports top-1 accuracy on seven benchmarks spanning images (CIFAR-10/100 (Krizhevsky et al., 2009), Tiny-ImageNet (Le & Yang, 2015), Caltech256 (Griffin et al., 2022), Food101 (Bossard

Table 3: Accuracy on various image and audio datasets. ProjNCE outperforms SupCon baseline methods for all dataset, although CE achieves the highest accuracy for SpeechCommands dataset.

| METHOD | CIFAR-10 | CIFAR-100 | TINY-IMAGENET | CALTECH256 | FOOD101 | CREMA-D | SPEECHCOMMANDS |
|---|---|---|---|---|---|---|---|
| SIMCLR | 74.37 | 49.51 | 40.23 | 46.01 | 40.76 | 50.32 | 29.40 |
| CE | 93.72 | 70.40 | 52.63 | 85.36 | 59.76 | 55.80 | **86.40** |
| SUPCON | 93.86 | 71.51 | 58.35 | 87.62 | 59.57 | 58.05 | 73.97 |
| PACO (CUI ET AL., 2021) | 94.05 | 71.94 | 57.05 | 77.64 | 60.31 | 59.47 | 71.94 |
| PROJNCE-PERP | **94.31** | 72.53 | 57.56 | 87.47 | 59.78 | **60.37** | 75.52 |
| PROJNCE-MLP | 94.17 | **72.90** | **59.01** | **88.57** | **60.71** | 58.25 | 73.07 |

Table 4: Performance of ProjNCE and baselines on the STL10 dataset with noisy labels. We report top-1 accuracy across various noise probabilities $p \in \{0, 0.1, \cdots, 0.7\}$. ProjNCE achieves best performance for all settings.

| CATEGORY | METHOD | $p = 0$ | $p = 0.1$ | $p = 0.2$ | $p = 0.3$ | $p = 0.4$ | $p = 0.5$ | $p = 0.6$ | $p = 0.7$ |
|---|---|---|---|---|---|---|---|---|---|
| CE | CE | 74.36 | 69.45 | 65.98 | 61.00 | 56.19 | 48.14 | 40.85 | 24.78 |
|  | SCE (WANG ET AL., 2019) | 73.38 | 70.29 | 67.34 | 61.64 | 54.48 | 49.88 | 40.69 | 26.93 |
| SSCL | SIMCLR (CHEN ET AL., 2020) | 74.40 | 70.64 | 67.60 | 62.31 | 54.93 | 50.51 | 41.93 | 35.54 |
|  | RINCE (CHUANG ET AL., 2022) | 20.16 | 22.36 | 23.33 | 21.28 | 23.18 | 21.36 | 21.26 | 21.26 |
| SUPCL | SUPCON (KHOSLA ET AL., 2020) | 79.80 | 75.05 | 70.46 | 63.39 | 56.32 | 47.31 | 42.86 | 26.23 |
|  | SYMNCE (CUI ET AL., 2025) | 44.40 | 50.08 | 47.56 | 49.54 | 45.46 | 37.66 | 28.51 | 18.94 |
|  | RINCE (CHUANG ET AL., 2022) | 32.16 | 31.36 | 28.40 | 24.89 | 25.45 | 25.66 | 22.80 | 27.66 |
|  | PROJNCE-MED (OURS) | 81.65 | **77.13** | 73.00 | 67.26 | 64.50 | 51.24 | 29.33 | 27.80 |
|  | PROJNCE-MLP (OURS) | **81.88** | 77.03 | **73.74** | **67.73** | **65.66** | **58.18** | **52.03** | **38.83** |

et al., 2014)) and audio (CREMA-D (Cao et al., 2014), SpeechCommands (Warden, 2018)). Across datasets except SpeechCommands, ProjNCE attains the best result. While CE yields the best performance for SpeechCommands, ProjNCE outperforms SupCon for all datasets, which validates the effectiveness of ProjNCE over SupCon. These results are consistent with our analysis: 1) ProjNCE preserves a valid mutual-information bound, which translates into higher classification accuracy across modalities; 2) a proper projection can boost accuracy and robustness.

Table 4 presents classification accuracies for noise rates $p$ on STL-10 (Coates et al., 2011). ProjNCE-MLP attains the highest accuracy in nearly all settings, except at $p = 0.1$, where ProjNCE-med slightly outperforms it. We attribute this to the MLP-based projection learning more robust class embeddings.

In Appendix D, we provide additional results—including feature-noise experiments—again showing that ProjNCE outperforms SupCon-based baselines. As discussed in Section 3 (see also Q2), SupCon's centroid-based label embedding is not universally optimal; thus, optimizing the projection is necessary. Sensitivity analyses of 1) embedding dimensionality in Table 6, 2) batch size in Table 7, and 3) temperature in Table 8 are provided in Appendix D, validating that the performance gain achieved by ProjNCE over SupCon is robust to hyperparameters. In terms of training stability, we track the classification accuracy (resp. mutual information) at every epoch in Figure 4 (resp. Figure 2). As ProjNCE refines SupCon with adjustment term and projection functions, it follows the behavior of SupCon.

Overall, these findings confirm that our proposed methods outperform SupCon in all settings, thereby establishing ProjNCE as a more robust foundation for supervised contrastive learning.

## 5 CONCLUSION

We introduced ProjNCE, a generalization of InfoNCE that subsumes SupCon and preserves a valid mutual-information lower bound via an adjustment term on negative pairs. By allowing flexible projection functions, ProjNCE delivers task-dependent improvements over rigid centroid embeddings. Extensive experiments on real-world image and audio datasets show that ProjNCE consistently outperforms SupCon. Moreover, ProjNCE integrates seamlessly into SupCon-style frameworks, providing a stronger foundation for future supervised contrastive learning methods.

A promising direction for future work is adapting ProjNCE to settings with auxiliary information beyond class labels. Thanks to its flexible projection design, ProjNCE can incorporate priors or side

information by instantiating $g_+$ and $g_-$ to encode class taxonomies, semantic label embeddings, or cost-sensitive relations—thereby emphasizing structure that the data alone may underrepresent.

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

## A  RELATED WORK

### A.1  CONTRASTIVE LEARNING

Self-supervised contrastive methods (Chen et al., 2020; Tian et al., 2020) learn an encoder $f : \mathcal{X} \to \mathcal{S}^{d_z - 1}$ by pulling semantically similar features closer and pushing dissimilar ones apart. Typically, each data sample is treated as its own class, with only its augmented version representing the same class. Various contrastive losses (Oord et al., 2018; Chen et al., 2020; Chuang et al., 2020; Barbano et al., 2022; Chopra et al., 2005; Hermans et al., 2017) have successfully produced robust representations for applications ranging from computer vision (Wang et al., 2021) to multimodal tasks (Radford et al., 2021; Girdhar et al., 2023). The InfoNCE loss (Oord et al., 2018) has been linked to mutual information (Poole et al., 2019), and subsequent analysis (Tschannen et al., 2019; Wang & Isola, 2020) has revealed how alignment–uniformity principles underlie its practical effectiveness. By contrast, supervised contrastive (SupCon) learning (Khosla et al., 2020) uses label information to group same-class features as positive pairs, often outperforming cross-entropy in downstream tasks.

### A.2  SUPCON AND ITS IMPROVEMENTS

While SSCL has been studied from an information-theoretic perspective, SupCon has not been explored in this way. Instead, the literature has proposed several enhancements to SupCon. First, SupCon can suffer from class collapse, where embeddings of the same class converge to a single point, increasing generalization error and reducing robustness (Graf et al., 2021). To mitigate this, various methods (Graf et al., 2021; Islam et al., 2021; Chen et al., 2022) introduce regularization or diversification strategies that preserve fine-grained information while maintaining strong discriminative performance.

Second, although robustness to label noise has been extensively studied for SSCL and cross-entropy approaches (Wang et al., 2019; Cui et al., 2023a; Xue et al., 2022; Chuang et al., 2022), only a few robust SupCon variants exist. Chuang et al. (Chuang et al., 2022) address noisy learning with RINCE, a drop-in replacement for InfoNCE that mitigates view noise. They link RINCE to symmetric losses used in noisy-label classification (Wang et al., 2019) and derive a mutual-information bound under the Wasserstein distance. Cui et al. (Cui et al., 2025) introduce an inclusive framework for robust contrastive techniques—such as nearest-neighbor sampling and RINCE—and propose SymNCE, a noise-robust adaptation of SupCon grounded in this theory.

UniCL (Yang et al., 2022) introduces an image–text–label space in which both image–label supervision and image–text pairs are trained with a single InfoNCE loss. By treating all views associated with the same label or caption as positives, UniCL unifies supervised and vision–language pretraining, but always within a standard InfoNCE form and without an explicit analysis of mutual-information bias in the supervised setting. X-Sample Contrastive Loss (Sobal et al., 2025) views contrastive learning as modifying a sample-similarity graph: instead of a binary positive/negative indicator, it uses a soft similarity matrix (e.g., from caption or taxonomy similarities) and trains the encoder to match these cross-sample affinities, yielding a weighted, graph-based generalization of InfoNCE. I-Con (Alshammari et al., 2025) takes an even broader perspective and shows that a wide range of representation-learning methods (clustering, dimensionality reduction, contrastive and supervised losses) can be written as minimizing an integrated KL divergence between a supervisory neighborhood distribution and the model's learned neighborhood distribution. ProjNCE is complementary to these efforts: rather than starting from a generic neighborhood or similarity graph, we focus specifically on the supervised contrastive setting and derive a projection-based InfoNCE formulation with an explicit adjustment term $R(X, C)$ (Eqs. (6)–(9)) that restores a valid MI lower bound for SupCon, together with concrete projection choices (orthogonal and median projections with a consistent estimator; Prop. 3.4) that can be instantiated and evaluated in practice.

Several recent methods also introduce explicit label embeddings into contrastive learning. Khalid et al. (2024) propose Label Supervised Contrastive Learning (LSCL) for imbalanced text classification, where each label is represented by a learnable prototype and the objective encourages text embeddings to be close to their label prototype and far from the others in either Euclidean or hyperbolic space. Lopez-Martin et al. (2022) propose similar idea to network intrusion detection and develop label-based and representative-label contrastive losses in which feature embeddings are contrasted against label prototypes or a single representative negative label. These approaches demonstrate the practical value of label prototypes in domain-specific settings, but they are formulated as purely supervised margin or softmax losses and do not provide an information-theoretic interpretation. In contrast, ProjNCE starts from a supervised InfoNCE/NWJ formulation of mutual information, explicitly derives the adjustment term $R(\mathbf{X}, C)$ that is missing in SupCon and prior label-prototype objectives, and frames label embeddings as projection functions $g_+$ and $g_-$ that can differ for positive and negative pairs. This projection-based view not only subsumes existing label-prototype methods as special cases but also leads to principled projection choices (orthogonal, median, and MLP) with desirable properties such as consistency and robustness.

## B  Proofs.

### B.1  Proof of Proposition 2.1

*Proof.* For any $\psi$ and $a \geq 0$, the multi-sample version of $I_{\mathrm{NWJ}}(X; Y)$ (Poole et al., 2019) is given by

$$
\begin{aligned}
I(\mathbf{X}; C) &\geq I_{\mathrm{NWJ}}(\mathbf{X}; C) \\
&= 1 + \mathbb{E}_{p(\boldsymbol{x}|\boldsymbol{c}_i) \prod_{j=1}^{N} p(\boldsymbol{c}_j)} \left[ \log \frac{e^{\psi(\boldsymbol{x},\boldsymbol{c}_i)}}{a} \right] - \mathbb{E}_{p(\boldsymbol{x}) \prod_{j=1}^{N} p(\boldsymbol{c}_j)} \left[ \frac{e^{\psi(\boldsymbol{x},\boldsymbol{c}_i)}}{a} \right].
\end{aligned}
\tag{19}
$$

Averaging $I_{\mathrm{NWJ}}(\mathbf{X}; C)$ over $i \in [N]$, we have

$$
\begin{aligned}
I_{\mathrm{NWJ}}(\mathbf{X}; C) = 1 &+ \frac{1}{N} \sum_{i=1}^{N} \mathbb{E}_{p(\boldsymbol{x}|\boldsymbol{c}_i) \prod_{j=1}^{N} p(\boldsymbol{c}_j)} \left[ \log \frac{e^{\psi(\boldsymbol{x},\boldsymbol{c}_i)}}{a} \right] \\
&- \frac{1}{N} \sum_{i=1}^{N} \mathbb{E}_{p(\boldsymbol{x}) \prod_{j=1}^{N} p(\boldsymbol{c}_j)} \left[ \frac{e^{\psi(\boldsymbol{x},\boldsymbol{c}_i)}}{a} \right].
\end{aligned}
\tag{20}
$$

Now, rewriting $\psi(\boldsymbol{x}, \boldsymbol{c}_i) = \psi(f(\boldsymbol{x}), g_+(\boldsymbol{c}_i))$ and setting $a = \frac{1}{N}\sum_{j=1}^{N} e^{\psi(f(\boldsymbol{x}), g_-(\boldsymbol{c}_j))}$, we obtain

$$
\begin{aligned}
I_{\mathrm{NWJ}}(\mathbf{X}; C) &= 1 + \frac{1}{N}\sum_{i=1}^{N} \mathbb{E}_{p(\boldsymbol{x}|\boldsymbol{c}_i)\prod_{j=1}^{N} p(\boldsymbol{c}_j)}\left[\log \frac{e^{\psi(f(\boldsymbol{x}), g_+(\boldsymbol{c}_i))}}{\frac{1}{N}\sum_{j=1}^{N} e^{\psi(f(\boldsymbol{x}), g_-(\boldsymbol{c}_j))}}\right] \\
&\quad - \frac{1}{N}\sum_{i=1}^{N} \mathbb{E}_{p(\boldsymbol{x})\prod_{j=1}^{N} p(\boldsymbol{c}_j)}\left[\frac{e^{\psi(f(\boldsymbol{x}), g_+(\boldsymbol{c}_i))}}{\frac{1}{N}\sum_{j=1}^{N} e^{\psi(f(\boldsymbol{x}), g_-(\boldsymbol{c}_j))}}\right] \\
&= 1 + \log N - I_{\mathrm{NCE}}^{\mathrm{self\text{-}p}}(\mathbf{X}; C) - \mathbb{E}_{p(\boldsymbol{x})\prod_{j=1}^{N} p(\boldsymbol{c}_j)}\left[\frac{\sum_{i=1}^{N} e^{\psi(f(\boldsymbol{x}), g_+(\boldsymbol{c}_i))}}{\sum_{j=1}^{N} e^{\psi(f(\boldsymbol{x}), g_-(\boldsymbol{c}_j))}}\right], \quad (21)
\end{aligned}
$$

which concludes the proof of Proposition 2.1. $\qquad\square$

### B.2 PROOF OF PROPOSITION 3.2

*Proof.* Proof of optimal critic of ProjNCE-perp leverages the optimal critic (Ma & Collins, 2018) of InfoNCE loss. Since ProjNCE-perp only differs in intermediate projections in the critic of InfoNCE, which does not affect the optimality, the optimal critic satisfies that

$$
\psi(f(\boldsymbol{x}), \overline{f}(\boldsymbol{c}))^\star \propto \log \frac{p(\boldsymbol{c}|\boldsymbol{x})}{p(\boldsymbol{c})} + \alpha(\boldsymbol{x}). \tag{22}
$$

Proof of the lower bound $I(\mathbf{X}; C) \geq I_{\mathrm{proj}}^{\mathrm{perp}}(\mathbf{X}; C) + \log N$. Similar to the proof of Proposition 2.1, we start from the multi-sample version of $I_{\mathrm{NWJ}}(X; Y)$ (Poole et al., 2019):

$$
\begin{aligned}
I(\mathbf{X}; C) &\geq I_{\mathrm{NWJ}}(\mathbf{X}; C) \\
&= 1 + \mathbb{E}_{p(\boldsymbol{x}|\boldsymbol{c}_i)\prod_{j=1}^{N} p(\boldsymbol{c}_j)}\left[\log \frac{e^{\psi(\boldsymbol{x}, \boldsymbol{c}_i)}}{a}\right] - \mathbb{E}_{p(\boldsymbol{x})\prod_{j=1}^{N} p(\boldsymbol{c}_j)}\left[\frac{e^{\psi(\boldsymbol{x}, \boldsymbol{c}_i)}}{a}\right]. \tag{23}
\end{aligned}
$$

Averaging $I_{\mathrm{NWJ}}(\mathbf{X}; C)$ over $i \in [N]$, rewriting $\psi(\boldsymbol{x}, \boldsymbol{c}_i) = \psi(f(\boldsymbol{x}), \overline{f}(\boldsymbol{c}_i))$ and setting $a = \frac{1}{N}\sum_{j=1}^{N} e^{\psi(f(\boldsymbol{x}), \overline{f}(\boldsymbol{c}_j))}$, we obtain

$$
\begin{aligned}
I_{\mathrm{NWJ}}(\mathbf{X}; C) &= 1 + \frac{1}{N}\sum_{i=1}^{N} \mathbb{E}_{p(\boldsymbol{x}|\boldsymbol{c}_i)\prod_{j=1}^{N} p(\boldsymbol{c}_j)}\left[\log \frac{e^{\psi(f(\boldsymbol{x}), \overline{f}(\boldsymbol{c}_i))}}{\frac{1}{N}\sum_{j=1}^{N} e^{\psi(f(\boldsymbol{x}), \overline{f}(\boldsymbol{c}_j))}}\right] \\
&\quad - \frac{1}{N}\sum_{i=1}^{N} \mathbb{E}_{p(\boldsymbol{x})\prod_{j=1}^{N} p(\boldsymbol{c}_j)}\left[\frac{e^{\psi(f(\boldsymbol{x}), \overline{f}(\boldsymbol{c}_i))}}{\frac{1}{N}\sum_{j=1}^{N} e^{\psi(f(\boldsymbol{x}), \overline{f}(\boldsymbol{c}_j))}}\right] \\
&= \frac{1}{N}\sum_{i=1}^{N} \mathbb{E}_{p(\boldsymbol{x}|\boldsymbol{c}_i)\prod_{j=1}^{N} p(\boldsymbol{c}_j)}\left[\log \frac{e^{\psi(f(\boldsymbol{x}), \overline{f}(\boldsymbol{c}_i))}}{\frac{1}{N}\sum_{j=1}^{N} e^{\psi(f(\boldsymbol{x}), \overline{f}(\boldsymbol{c}_j))}}\right] \\
&= \frac{1}{N}\sum_{i=1}^{N} \mathbb{E}_{p(\boldsymbol{x}|\boldsymbol{c}_i)\prod_{j=1}^{N} p(\boldsymbol{c}_j)}\left[\log \frac{e^{\psi(f(\boldsymbol{x}), \overline{f}(\boldsymbol{c}_i))}}{\sum_{j=1}^{N} e^{\psi(f(\boldsymbol{x}), \overline{f}(\boldsymbol{c}_j))}}\right] + \log N \\
&= -I_{\mathrm{proj}}^{\mathrm{perp}}(\mathbf{X}; C) + \log N. \tag{24}
\end{aligned}
$$

Substituting (24) into (23) gives

$$
I(\mathbf{X}; C) \geq -I_{\mathrm{proj}}^{\mathrm{perp}}(\mathbf{X}; C) + \log N. \tag{25}
$$

With an optimal critic $\psi(f(\boldsymbol{x}), \overline{f}(\boldsymbol{c}))^\star = \log \frac{p(\boldsymbol{c}|\boldsymbol{x})}{p(\boldsymbol{c})}$, $i$-th ProjNCE-perp for $\boldsymbol{x}_i$ can be written as

$$
I_{\mathrm{proj},i}^{\mathrm{perp}}(\mathbf{X}; C) - \log N = -\mathbb{E}_{p(\boldsymbol{x}_i|\boldsymbol{c}_i)\prod_{j=1}^{N} p(\boldsymbol{c}_j)}\left[\log \frac{\frac{p(\boldsymbol{c}_i|\boldsymbol{x}_i)}{p(\boldsymbol{c}_i)}}{\frac{1}{N}\sum_{j=1}^{N} \frac{p(\boldsymbol{c}_j|\boldsymbol{x}_i)}{p(\boldsymbol{c}_j)}}\right]. \tag{26}
$$

Taking $N \to \infty$, due to the strong law of large numbers, it follows that

$$\log N - I_{\text{proj,i}}^{\text{perp}}(\mathbf{X}; C) \stackrel{N \to \infty}{=} \mathbb{E}_{p(\boldsymbol{x}_i, \boldsymbol{c}_i)} \left[ \log \frac{\frac{p(\boldsymbol{c}_i|\boldsymbol{x}_i)}{p(\boldsymbol{c}_i)}}{\mathbb{E}_{p(\boldsymbol{c})}\left[\frac{p(\boldsymbol{c}|\boldsymbol{x}_i)}{p(\boldsymbol{c})}\right]} \right]$$

$$= \mathbb{E}_{p(\boldsymbol{x}_i, \boldsymbol{c}_i)} \left[ \log \frac{p(\boldsymbol{c}_i|\boldsymbol{x}_i)}{p(\boldsymbol{c}_i)} \right]$$

$$= I(\mathbf{X}; C). \tag{27}$$

This concludes the proof of Proposition 3.2. □

### B.3 PROOF OF PROPOSITION 3.3

*Proof.* According to (Ishida et al., 2022; Jeong et al., 2023), the categorical labels can be decomposed into soft label (i.e., posterior probability $p(\boldsymbol{c}|\boldsymbol{x})$) and some noise having zero-mean. Specifically, we can write that

$$\mathbb{1}\{C = \boldsymbol{c}\} = p(\boldsymbol{c}|\boldsymbol{z}) + \epsilon, \tag{28}$$

where $\mathbb{E}[\epsilon] = 0$. This model allows us to leverage results of non-parametric kernel estimator:

**Lemma B.1** (Corollary 4.3, (Ferraty & Vieu, 2004)). *Assume that:*

1. *$\mathcal{D}$ consists of independent samples.*

2. *Kernel $K$ has support $[0,1]$, is strictly decreasing, and is Lipschitz continuous;*

3. *$\exists \theta, \forall t \in [0,1], -K'(t) > \theta > 0$;*

4. *Hölder condition: there exist $C < \infty$ and $\beta > 0$, such that for all $(\boldsymbol{u}, \boldsymbol{v}) \in \mathcal{S}^{d_z - 1}$, $|\text{NW}_h(\boldsymbol{c}|\mathbf{u}; \mathcal{D}) - \text{NW}_h(\boldsymbol{c}|\mathbf{v}; \mathcal{D})| \le Cd(\mathbf{u}, \mathbf{v})$, where $d$ is the metric used in the kernel;*

5. *$\exists p \ge 2, \mathbb{E}[p(\boldsymbol{c}|\boldsymbol{z})]^p < \infty$;*

6. *$\sup_{\mathbf{u}, \mathbf{v}} \mathbb{E}[|p(\boldsymbol{c}|\mathbf{u})p(\boldsymbol{c}|\mathbf{v})| \mid \mathbf{u}, \mathbf{v}] < \infty$;*

7. *There exists $\kappa > 0$ such that $\inf_{\boldsymbol{z} \in \mathcal{S}^{d_z - 1}} p_{\boldsymbol{z}}(\boldsymbol{z}) \ge \kappa$, where $p_{\boldsymbol{z}} = \frac{dF_{\boldsymbol{z}}}{d\mu}$ is the Radon-Nikodym derivative of the distribution function $F_{\boldsymbol{z}}$ with respect to the Lebesgue measure $\mu$ on $\mathcal{S}^{d_z - 1}$.*

*Then, it holds that*

$$\sup_{\boldsymbol{z} \in \mathcal{S}^{d_z - 1}} |p(\boldsymbol{c}|\boldsymbol{z}) - \text{NW}_h(\boldsymbol{c}|\boldsymbol{z}, \mathcal{D})| = O(h^\beta) + O\left(\sqrt{\frac{\ln N}{h^{d_z} N}}\right), a.s. \tag{29}$$

The assumption 1 holds as $\mathcal{D}$ consists of i.i.d. samples. The NW estimator in (16) uses a kernel satisfying the assumption 2 and 3 in Lemme B.1. The assumption 5 is true since $\mathbb{E}[p(\boldsymbol{c}|\boldsymbol{z})]^p \le 1 < \infty$. The assumption 6 also holds because $\sup_{\mathbf{u}, \mathbf{v}} \mathbb{E}[|p(\boldsymbol{c}|\mathbf{u})p(\boldsymbol{c}|\mathbf{v})| \mid \mathbf{u}, \mathbf{v}] \le 1 < \infty$. Now, let us assume that the assumptions 4 and 7 are satisfied. Then, as $N \to \infty$ and $h \to 0$ with $\sqrt{\frac{\ln N}{h^{d_z} N}} \to 0$, it follows that

$$\text{NW}_h(\boldsymbol{c}|\boldsymbol{z}, \mathcal{D}) \stackrel{a.s.}{\to} p(\boldsymbol{c}|\boldsymbol{z}). \tag{30}$$

With (30) and the law of large numbers, $\hat{f}$ in (17) converges as

$$\hat{f}(\boldsymbol{c}) \to \frac{\mathbb{E}_{p(\boldsymbol{x})}[p(\boldsymbol{c}|f(\boldsymbol{x}))f(\boldsymbol{x})]}{\mathbb{E}_{p(\boldsymbol{x})}[p(\boldsymbol{c}|f(\boldsymbol{x}))]}$$

$$= \overline{f}(\boldsymbol{c}). \tag{31}$$

This concludes the proof of Proposition 3.3 □

## C   DEFINITION

**Definition C.1** (ProjNCE-perp). ProjNCE-perp loss is defined as

$$I_{\text{proj}}^{\text{perp}} = \frac{1}{N} \sum_{i=1}^{N} \mathbb{E}_P \left[ -\log \frac{e^{\psi(f(\boldsymbol{x}_i), \overline{f}(\boldsymbol{c}_i))}}{\sum_{j=1}^{N} e^{\psi(f(\boldsymbol{x}_i), \overline{f}(\boldsymbol{c}_j))}} \right], \tag{32}$$

where $\overline{f}(\boldsymbol{c}) = \mathbb{E}[f(\mathbf{X})|C = \boldsymbol{c}]$.

**Definition C.2** (ProjNCE-med). ProjNCE-med loss is defined as

$$I_{\text{proj}}^{\text{med}} = \frac{1}{N} \sum_{i=1}^{N} \mathbb{E}_P \left[ -\log \frac{e^{\psi(f(\boldsymbol{x}_i), f_{\text{med}}(\boldsymbol{c}_i))}}{\sum_{j=1}^{N} e^{\psi(f(\boldsymbol{x}_i), f_{\text{med}}(\boldsymbol{c}_j))}} \right], \tag{33}$$

where $f_{\text{med}}(\boldsymbol{c}_i) = \text{median}(\{f(\boldsymbol{x}_p)\}_{p \in \mathcal{P}(i)})$.

**Definition C.3** (ProjNCE-MLP). Let $\mathcal{F} : \mathcal{C} \to \mathcal{Z}$ be an arbitrary MLP model. Then, ProjNCE-MLP loss is defined as

$$I_{\text{proj}}^{\text{MLP}} = \frac{1}{N} \sum_{i=1}^{N} \mathbb{E}_P \left[ -\log \frac{e^{\psi(f(\boldsymbol{x}_i), \mathcal{F}(\boldsymbol{c}_i))}}{\sum_{j=1}^{N} e^{\psi(f(\boldsymbol{x}_i), \mathcal{F}(\boldsymbol{c}_j))}} \right]. \tag{34}$$

## D   EXPERIMENTS

We evaluate ProjNCE with several projection choices against standard baselines. For images, we use a ResNet-18 encoder (He et al., 2016); for audio, we use a 3-layer 2-D CNN. Both encoders produce 128-dimensional embeddings. For ProjNCE-perp, we employ a learnable matrix of size $128 \times |\mathcal{C}|$ that linearly maps each class label into the embedding space. For audio inputs, raw waveforms are converted to Mel spectrograms of size $(1, 128, 204)$ or $(1, 128, 32)$, depending on the dataset. We train encoders with AdamW (Loshchilov, 2017) using batch size 256, a total of 500 epochs (100 for ImageNet (Deng et al., 2009) and 150 for SpeechCommands (Warden, 2018)), learning rate 0.01, weight decay $10^{-4}$, and temperature 0.07 for all contrastive losses. Data augmentation and contrastive-learning settings follow Khosla et al. (2020), which builds on SimCLR (Chen et al., 2020).

**Classification for the general case.**   To obtain accuracy from the learned embeddings, we use a zero-shot classifier. For each class, we compute a class embedding in the shared space; a test sample is assigned the class with the highest cosine similarity between its class embedding and the sample's embedding. We evaluate zero-shot accuracy at every training epoch and report the best epoch for noiseless experiments.

**Classification with noisy samples.**   With noisy labels or noisy features, accuracy fluctuates substantially in early epochs, making robust comparisons difficult. Accordingly, for noisy settings we report accuracy at the final epoch, after performance stabilizes (typically within a few hundred epochs). Optimization hyperparameters match those above, except we train the probe for 100 epochs.

### D.1   KERNEL AND BANDWIDTH

Figure 3 plots CIFAR-10 and STL10 accuracy versus bandwidth $h$ for ProjNCE-perp using $\ell_1$, $\ell_2$, and cosine dissimilarities. Accuracy is broadly similar across metrics—consistent with Proposition 3.3—with a single anomaly at $(h = 0.7, d = \cos)$ on STL10. This pattern indicates that ProjNCE-perp requires only minimal kernel-parameter tuning; in practice, evaluating small grid of bandwidth values typically suffices.

Throughout all experiments we use the Epanechnikov (parabolic) kernel $K(t) \propto (1 - t^2)_+$ in the Nadaraya–Watson estimator, applied to scaled distances $d(z, z')/h$. We chose this kernel because it is known to be optimal in a mean-squared-error sense among symmetric kernels with compact support, and it naturally emphasizes local neighbors while smoothly down-weighting more distant points. Classical nonparametric regression theory further suggests that, under mild conditions, the choice of kernel has only a second-order effect compared to the bandwidth. Our empirical curves in Figure 3 are consistent with this view: performance is stable across distance metrics, and the main practical consideration is selecting a reasonable bandwidth $h$, rather than fine-tuning the specific kernel shape.

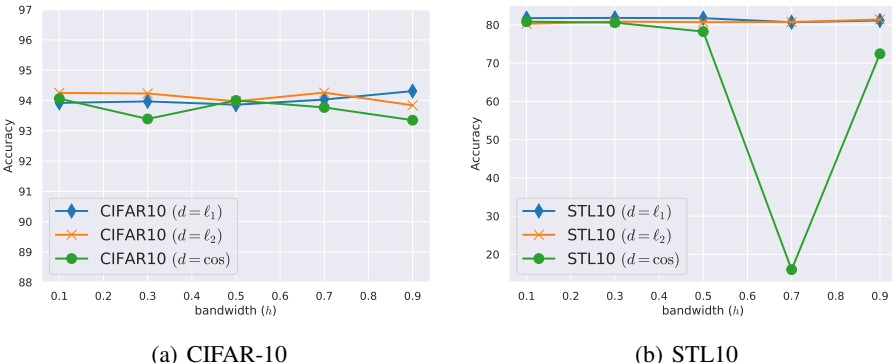

(a) CIFAR-10          (b) STL10

Figure 3: Effect of bandwidth $h$. Results are shown for $\ell_1, \ell_2$, and cosine (cos) dissimilarities. With the exception of $(d = cos, h = 0.7)$ on STL10, ProjNCE-perp is largely insensitive to kernel parameters. The sharp dip at $(d = cos, h = 0.7)$—with accuracy $< 20\%$—likely reflects an outlier or convergence to a poor local optimum. Overall, ProjNCE-perp requires minimal hyperparameter tuning.

## D.2   SENSITIVITY ANALYSIS OF $\beta$ IN (12)

In Figure 1, we visualize embeddings of SupCon, ProjNCE, and (12) with $\beta \in \{5, 10\}$ to see how the adjustment term $R(\mathbf{X}; C)$ affects the learned embeddings. The embedding visualization shows that $R(\mathbf{X}; C)$ makes the embeddings being spread out. The corresponding accuracy for $\mathcal{L}$ (12) with different $\beta$ is reported in Table 5. For CIFAR-10, CIFAR-100, and Tiny-imagenet, setting $\beta = 1$ (equivalent to ProjNCE) achieves the best accuracy, especially it outperforms SupCon. This result aligns with our analysis in Section 2.2, specifically with Corollary 2.2 that shows a valid mutual information bound is obtained by adding $R(\mathbf{X}; C)$ to SupCon. We expect that the performance gain stems from such a theoretically correct mutual information bound.

Table 5: Classification accuracies with various $\beta$ in (12). As expected from Corollary 2.2, ProjNCE with $\beta = 1$ yields the best performance.

| METHOD | CIFAR-10 | CIFAR-100 | TINY-IMAGENET |
|---|---|---|---|
| SUPCON ($\beta = 0$) | 93.86 | 71.51 | 58.50 |
| PROJNCE ($\beta = 1$) | **94.20** | **71.84** | **58.98** |
| PROJNCE ($\beta = 5$) | 93.66 | 67.26 | 44.91 |
| PROJNCE ($\beta = 10$) | 93.20 | 42.68 | 17.68 |

## D.3   SENSITIVITY ANALYSIS OF EMBEDDING DIMENSION

Table 6 reports the accuracies when varying the embedding dimension $d \in \{16, 32, 64, 128, 256\}$ for SupCon and ProjNCE. Overall, both methods are fairly insensitive to the choice of $d$: performance on all three datasets fluctuates within roughly one percentage point across the tested dimensions. Importantly, ProjNCE consistently matches or outperforms SupCon for almost all configurations.

On CIFAR-10, ProjNCE attains the best accuracy at $d = 128$ (94.20%), slightly improving over the best SupCon setting (94.11% at $d = 16$). On CIFAR-100, ProjNCE reaches its peak at $d = 16$ (72.67%), again marginally higher than the best SupCon result (72.48% at $d = 16$). For Tiny-ImageNet, ProjNCE achieves the highest accuracy of 58.98% at $d = 128$, exceeding the best SupCon configuration (58.56% at $d = 256$). These results indicate that the gains of ProjNCE over SupCon are not tied to a particular embedding dimension and that ProjNCE remains competitive even in regimes with very low-dimensional embeddings. In the main experiments we therefore adopt $d = 128$ as a default choice, which offers a good trade-off between accuracy and representation size across datasets.

Table 6: Classification accuracies with various embedding dimension $d \in \{16, 32, 64, 128, 256\}$.

| METHOD | CIFAR-10 | CIFAR-100 | TINY-IMAGENET |
|---|---|---|---|
| SUPCON ($d = 16$) | **94.11** | **72.48** | 56.52 |
| SUPCON ($d = 32$) | 94.09 | 72.03 | 57.88 |
| SUPCON ($d = 64$) | 93.96 | 72.08 | 58.32 |
| SUPCON ($d = 128$) | 93.86 | 71.51 | 58.50 |
| SUPCON ($d = 256$) | 93.44 | 71.62 | **58.56** |
| PROJNCE ($d = 16$) | 93.88 | **72.67** | 58.14 |
| PROJNCE ($d = 32$) | 93.96 | 72.24 | 58.65 |
| PROJNCE ($d = 64$) | 94.04 | 72.39 | - |
| PROJNCE ($d = 128$) | **94.20** | 71.84 | **58.98** |
| PROJNCE ($d = 256$) | 94.07 | 72.09 | 58.70 |

## D.4 DIFFERENT BATCH SIZE

Table 7: Classification accuracies with different batch size $B \in \{128, 256, 512, 1024\}$.

| METHOD | CIFAR-10 | CIFAR-100 | TINY-IMAGENET |
|---|---|---|---|
| SUPCON ($B = 128$) | 93.85 | 71.30 | 58.35 |
| SUPCON ($B = 256$) | 93.86 | 71.51 | **58.50** |
| SUPCON ($B = 512$) | **94.03** | 71.56 | 57.16 |
| SUPCON ($B = 1024$) | 93.79 | **71.86** | 57.10 |
| PROJNCE ($B = 128$) | 93.91 | **72.94** | 53.51 |
| PROJNCE ($B = 256$) | 94.20 | 71.84 | **58.98** |
| PROJNCE ($B = 512$) | **94.26** | 72.29 | 57.93 |
| PROJNCE ($B = 1024$) | 93.69 | 72.26 | 57.93 |

Table 7 summarizes the effect of the batch size $B \in \{128, 256, 512, 1024\}$ on classification accuracy. Overall, both SupCon and ProjNCE exhibit only moderate sensitivity to $B$, with variation within roughly one percentage point on each dataset. Across all batch sizes, ProjNCE matches or improves upon SupCon on CIFAR-10 and CIFAR-100, and is competitive on Tiny-ImageNet once $B \geq 256$.

On CIFAR-10, the best SupCon configuration reaches 94.03% at $B = 512$, while ProjNCE attains a slightly higher 94.26% at the same batch size. On CIFAR-100, ProjNCE consistently outperforms SupCon for all tested $B$, peaking at 72.94% with $B = 128$ compared to SupCon's 71.86% at $B = 1024$. Tiny-ImageNet is more sensitive to small batch sizes for ProjNCE—the adjustment term is harder to estimate with limited negatives and performance drops at $B = 128$—but for $B \geq 256$ ProjNCE again dominates, achieving the highest accuracy of 58.98% at $B = 256$ versus 58.50% for SupCon. These results suggest that ProjNCE benefits from sufficiently large batches, as expected for contrastive objectives, but its gains over SupCon are robust across a wide range of practical batch sizes.

## D.5 SENSITIVITY ANALYSIS OF TEMPERATURE

Table 8: Classification accuracies with different temperature $\tau \in \{0.07, 0.3, 0.7\}$.

| METHOD | CIFAR-10 | CIFAR-100 | TINY-IMAGENET |
|---|---|---|---|
| SUPCON ($\tau = 0.07$) | **93.86** | **71.51** | **58.50** |
| SUPCON ($\tau = 0.3$) | 93.78 | 67.10 | 50.13 |
| SUPCON ($\tau = 0.7$) | 91.98 | 47.25 | 35.25 |
| PROJNCE ($\tau = 0.07$) | **94.20** | **71.84** | **58.98** |
| PROJNCE ($\tau = 0.3$) | 93.74 | 66.13 | 50.57 |
| PROJNCE ($\tau = 0.7$) | 92.27 | 48.60 | 34.31 |

Table 7 shows the effect of varying the temperature $\tau \in \{0.07, 0.3, 0.7\}$ on classification accuracy. As expected for contrastive objectives, both SupCon and ProjNCE are quite sensitive to this hyperparameter: performance degrades as the temperature increases and the softmax over similarities becomes flatter. The smallest temperature $\tau = 0.07$ yields the best results for all datasets and both methods.

At this default temperature, ProjNCE consistently outperforms SupCon, achieving 94.20% vs. 93.86% on CIFAR-10, 71.84% vs. 71.51% on CIFAR-100, and 58.98% vs. 58.50% on Tiny-ImageNet. For larger temperatures ($\tau = 0.3$ and $\tau = 0.7$), the accuracies of both methods drop substantially—especially on CIFAR-100 and Tiny-ImageNet—but ProjNCE closely tracks SupCon and remains slightly better on most dataset–temperature combinations. These results indicate that ProjNCE inherits the usual temperature sensitivity of contrastive learning, but its advantage over SupCon is robust around the standard setting $\tau = 0.07$, which we adopt for all main experiments.

### D.6 Training Stability

To check whether the additional projection and adjustment terms introduce optimization difficulties, we monitor the top-1 validation accuracy during training. Figure 4 reports the accuracy of CE, PaCo, SupCon, and the three ProjNCE variants (ProjNCE, ProjNCE-MLP, and ProjNCE-perp) over 500 epochs on CIFAR-10, CIFAR-100, Tiny-ImageNet, and CREMA-D. Across all datasets, the contrastive methods exhibit smooth and monotone learning curves: after an initial warm-up period, accuracies steadily improve and then plateau without signs of divergence or collapse.

The three ProjNCE variants closely track SupCon throughout training and consistently converge to slightly higher accuracies, indicating that the ProjNCE objective is as easy to optimize as SupCon. In particular, the variance of ProjNCE and SupCon trajectories is comparable on all four datasets, while CE tends to show larger fluctuations and slower convergence, especially on the audio dataset CREMA-D. Overall, these curves suggest that ProjNCE maintains the favorable training stability of standard supervised contrastive learning while providing improved final performance.

### D.7 Classification with Noisy Features

Table 9: Performance on the CIFAR-10 dataset with noisy features. Each image (with pixel value from 0 to 255) is corrupted by adding Gaussian noise with zero mean and 70 standard deviation. We report top-1 accuracy. **Boldface** indicates the best accuracy for each category.

| Category | Method | CIFAR-10 with noisy image |
|---|---|---|
| CE | CE | 57.86 |
| | SCE (Wang et al., 2019) | **63.59** |
| SSCL | SimCLR (Chen et al., 2020) | **46.49** |
| | RINCE (SSCL) (Chuang et al., 2022) | 39.46 |
| SupCL | SupCon (Khosla et al., 2020) | 60.04 |
| | SymNCE (Cui et al., 2025) | 60.24 |
| | RINCE (SupCL) (Chuang et al., 2022) | 52.60 |
| | ProjNCE-perp (Ours) | 61.55 |
| | ProjNCE-med (Ours) | **62.48** |
| | ProjNCE-MLP (Ours) | 60.03 |
| Hybrid | JointTraining (Cui et al., 2023a) with SupCon | **56.25** |
| | JointTraining (Cui et al., 2023a) with ProjNCE | 56.07 |

In addition to the noisy-label experiments, we assess robustness to noisy features by corrupting the CIFAR-10 dataset with additive Gaussian noise (zero mean, standard deviation 70) applied to each pixel value in the range 0–255.

Table 9 reports top-1 accuracy for each method, including cross-entropy (CE)-based robust baselines (SCE (Wang et al., 2019)), SupCon-based robust baselines (both self-supervised and supervised version of RINCE (Chuang et al., 2022), SymNCE (Cui et al., 2025)), and JointTraining (Cui et al., 2023a). Under noisy features, robust CE-based method, SCE, achieves the highest accuracy mainly due to a mild usage of noisy feature in CE method, while contrastive learning mainly exploits

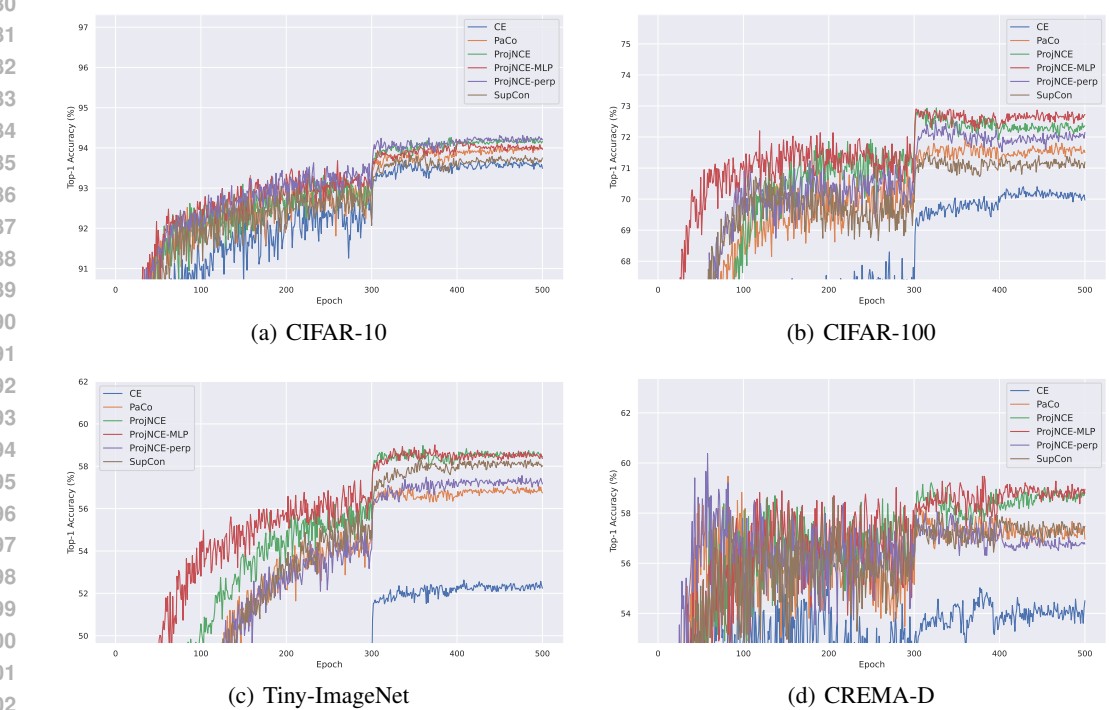

Figure 4: Top-1 validation accuracy of CE, PaCo, SupCon, and our ProjNCE variants (ProjNCE, ProjNCE-MLP, ProjNCE-perp) over 500 training epochs on (a) CIFAR-10, (b) CIFAR-100, (c) Tiny-ImageNet, and (d) CREMA-D. ProjNCE follows similarly stable optimization trajectories as SupCon and PaCo and consistently converges to slightly higher accuracies, indicating that the proposed objective does not compromise training stability.

features that are perturbed for classification. Among SupCon baselines, ProjNCE-med attains the best performance, surpassing both SymNCE and RINCE.

