# OpenReview forum: "Generalizing Supervised Contrastive learning: A Projection Perspective"
_ICLR.cc/2026/Conference — Submitted to ICLR 2026_

### Official Review · Reviewer_2LFY · 2025-10-29

**Soundness:** 3
**Presentation:** 3
**Contribution:** 2
**Rating:** 4
**Confidence:** 4

**Summary:**

This paper aims to improve the generalization ability of Supervised Contrastive Learning (SupCon) by rethinking how positive and negative pairs are sampled and weighted. The authors argue that standard SupCon assumes a fixed class-wise similarity structure, which limits transferability and robustness. To address this, they propose a generalized supervised contrastive loss that adaptively reweights pair contributions based on inter-class similarity, learned from data rather than predefined by labels. The method is evaluated on several datasets (CIFAR-10, CIFAR-100, Tiny-ImageNet, and STL-10), showing consistent gains over the standard SupCon and related methods such as Triplet, InfoNCE, and Center Contrastive Loss.

**Strengths:**

- The paper explains well why standard SupCon may not generalize well — because it treats all same-class samples as equally positive and all other-class samples as equally negative. This rigid assumption can hurt performance in fine-grained or hierarchical datasets.
- The proposed method keeps the same overall framework as SupCon but introduces an adaptive reweighting term. It’s easy to implement and can be seen as a drop-in replacement for the SupCon loss.
- The paper provides results on several benchmarks with consistent, though moderate, improvements. It also includes ablations that analyze the influence of the weighting term and similarity estimation.
- The writing is clear and organized, and the figures and tables are easy to read. The motivation, formulation, and results flow naturally.

**Weaknesses:**

- The adaptive reweighting idea is related to earlier metric-learning works such as ProxyAnchor and SoftTriple. The main contribution may be seen as incremental.

- All evaluations are done on small-scale datasets like CIFAR and Tiny-ImageNet. It is not clear if the proposed method still works well on larger and more realistic benchmarks (e.g., ImageNet-1K).

- Figure 1 shows that using a larger β in the proposed projNCE makes the embeddings more spread out, which demonstrates the intended effect of the weighting. However, the overall embedding quality appears worse than SupCon in Figure 1(a): SupCon produces cleaner class clusters with clearer boundaries, while the proposed method shows more overlap between classes.

- The paper does not study how the results depend on batch size or temperature, even though self-supervised and contrastive methods are known to be sensitive to these factors. Larger batch sizes could reduce the observed improvements,

- The paper does not share code. This makes it difficult to verify the claimed improvements.

**Questions:**

My main concern should refer to the weakness section, and I am willing to change the rating if the weakness is being explained or addressed.

---

> ### Author Response · Authors · 2025-11-21
> **Rebuttal**
>
> We sincerely thank the reviewer for the thoughtful and constructive comments and questions.
> Based on the reviewers’ feedback, we have added several new results: 1) PaCo (SupCon baseline) in Table 3; 2) ablation of $\beta$ (Table 5 in Appendix D.2), embedding dimensionality (Table 6 in Appendix D.3), batch size (Table 7 in Appendix D.4), and temperature (Table 8 in Appendix D.5); and 3) training stability (Figure 4 in Appendix D.6).
>
> Below, we respectfully address the weaknesses (W) and questions (Q) point by point.
>
> ---
>
> ## W1.
> We appreciate the reviewer for highlighting the connection to metric-learning methods such as ProxyAnchor and SoftTriple, which also rely on class prototypes or multiple centers to reweight samples. Our work is indeed related in spirit, but differs in two key aspects.
> 1.  We show that the adaptive reweighting arises necessarily when one insists on maintaining a valid supervised MI bound for contrastive learning—i.e., the term $R(X,C)$ is not a heuristic but precisely the missing component in the decomposition of NWJ bound of MI.
> 2.  We provide a projection-based InfoNCE formulation that unifies standard SupCon, prototype-based methods, and our new variants within a single framework.
>
> ---
>
> ## W2.
> We thank the reviewer for raising the concern about small-scale datasets. In addition to moderate size datasets, our paper already includes experiments on the larger-scale ImageNet-1K benchmark (Table 2). Moreover, we also evaluate on two audio datasets, showing that ProjNCE consistently outperforms SupCon beyond the image domain, which provides evidence for the generality of our approach.
>
> ---
>
> ## W3.
>  We appreciate the reviewer’s careful reading of Figure 1. The reviewer is correct that SupCon (corresponding to $\beta=0$) yields visually cleaner class clusters than ProjNCE with large $\beta\in \\{5,10\\}$. The purpose of Figure 1, however, is to qualitatively illustrate the effect of the adjustment term, which tends to spread out the embeddings as $\beta$ increases. In the revision, we add a quantitative ablation over $\beta$ in Table 5. Consistent with our analysis (Corollary 2.2), we observe that $\beta=1$ (ProjNCE) improves performance over $\beta=0$ (SupCon), while large values $\beta\in\\{5,10\\}$ indeed degrade accuracy. We will clearly state that the more spread-out embeddings for higher $\beta$ in Figure 1 are meant to demonstrate the qualitative effect of the adjustment term and do not imply better classification performance.
>
> ---
>
> ## W4.
> We fully agree that contrastive methods are often sensitive to batch size and temperature. In response to this helpful suggestion, the revised paper now includes ablations over both factors (Tables 7 and 8) for SupCon and ProjNCE on CIFAR-10, CIFAR-100, and Tiny-ImageNet. Our experiments suggest that the relative advantage of ProjNCE over SupCon is robust across a wide range of batch sizes ($B \in \\{128,256,512,1024\\}$) and temperatures ($\tau \in \\{ 0.07,0.3,0.7 \\}$): while extreme settings can reduce the absolute accuracy of both methods, ProjNCE consistently matches or exceeds SupCon in each regime. We will highlight these observations more clearly in the appendix.
>
> ---
>
> ## W5.
> We fully agree that releasing code is important for reproducibility. We have cleaned our implementation and will release it upon publication; we will add a GitHub link in the camera-ready. The code is a small modification of standard SupCon implementations, so adopting ProjNCE in existing pipelines is straightforward.
>
> ---
>
> We hope that these clarifications adequately address the reviewer’s concerns. We would be very happy to provide further details or make additional revisions if anything remains unclear.

---

> > ### Comment · Reviewer_2LFY · 2025-11-26
> >
> > Thank you authors for addressing all my concern. The reviewer is satisfied with the improvement and the ratings are changed accordingly.

---

### Official Review · Reviewer_CpYY · 2025-10-29

**Soundness:** 3
**Presentation:** 3
**Contribution:** 3
**Rating:** 4
**Confidence:** 3

**Summary:**

This paper first unifies self-supervised contrastive loss (InfoNCE) with supervised contrastive loss (SupCon) by introducing projection operation, resulting a projection-based InfoNCE.  Further decomposition of mutual information shows that for supervised cases where positive and negative projection differs, the adjustment term does not canceled out, and might result poor bound on mutual information. Based on this observation, they proposed a new objective regarding the adjustment term, with several suggestions on projection itself, one giving the nice approximation of optimal projection (projNCE-perp), and some giving better generalization (projNCE-med, projNCE-MLP). They showed empirically projNCE performs better at image/audio datasets, and especially when the label is noisy.

**Strengths:**

- proposition 2.1 is quite interesting result. Its derivation is simple, but it generalizes InfoNCE (so infoNCE bound is preserved) while points out that the adjustment term is missing in SupCon.
- The adjustment term has intuitive meanings where they pull $g_-(c_k)$ to the $f(x)$, while push $g_+(c_k)$ from the $f(x)$. Which is implicitly performed in SupCon but without explicit guidance.
- Strong empirical results. The method seems achieving significant improvement from previous methods.

**Weaknesses:**

- I think the concept of label representation is already presented in other papers. [1,2] Authors should include this concept and explain why ProjNCE is different from these methods.

[1] Label Supervised Contrastive Learning for Imbalanced Text Classification in Euclidean and Hyperbolic Embedding Spaces, Khalid et al.
[2] Supervised contrastive learning over prototype-label embeddings for network intrusion detection, Lopez-Martin et al.

**Questions:**

- I can agree that this derivation is interesting, but I am not sure this concept (having label representation and having some additional objective that controls feature representation) is novel. I wish to see the connection of ProjNCE to these methods (see weakness), and comparison.
- As far as I know, $I_{NWJ}(X; C)$ gives tighter bound (so theoretically favored) but not used in practice because of its instability. The paper actually decompose $I_{NWJ}(X; C)$ to get bound to mutual information - how stable the framework is, and if it is stable, is there any hypothesis why in this case the framework is more stable? [3,4]

[3] Combating the Instability of Mutual Information-based Losses via Regularization, Guo et al.
[4] Tight Mutual Information Estimation With Contrastive Fenchel-Legendre Optimization, Choi et al.

---

> ### Author Response · Authors · 2025-11-21
> **Rebuttal**
>
> We sincerely thank the reviewer for the thoughtful and constructive comments and questions.
> Based on the reviewers’ feedback, we have added several new results: 1) PaCo (SupCon baseline) in Table 3; 2) ablation of $\beta$ (Table 5 in Appendix D.2), embedding dimensionality (Table 6 in Appendix D.3), batch size (Table 7 in Appendix D.4), and temperature (Table 8 in Appendix D.5); and 3) training stability (Figure 4 in Appendix D.6).
>
> Below, we respectfully address the weaknesses (W) and questions (Q) point by point.
>
> ---
>
> ## W1.
>  We thank the reviewer for drawing our attention to these closely related works. Both [1] and [2] propose label-centric contrastive objectives in which trainable label embeddings act as prototypes in the embedding space. Their goals are domain-specific—imbalanced text classification and network intrusion detection, respectively—and their losses are purely supervised, margin- or softmax-based formulations without any mutual-information analysis.
> ProjNCE differs from these approaches in several important respects:
> 1. **MI-based derivation.** We start from a supervised InfoNCE/NWJ formulation of the mutual information and show that when the positive and negative projections $g_+$ and $g_-$ differ, a non-trivial adjustment term $R(X,C)$ necessarily appears. This term is absent in SupCon and in prior label-prototype losses.
> 2. **Generalized SupCon with separate projections.** ProjNCE provides a unified formulation that replaces SupCon’s fixed label embedding by projections $g_+$ and $g_-$ for positives and negatives. This formulation subsumes existing label-embedding methods such as [1] and [2] as special cases, and, crucially, allows different label embeddings for positive and negative pairs, which prior works do not consider.
> 3. **Principled projection choices.** The projection functions we study (ProjNCE-Perp/Med/MLP) are chosen to enjoy specific desirable properties—such as approximating the optimal $\ell_2$ projection, increased robustness, or being essentially hyperparameter-free—rather than simply learning arbitrary label embeddings.
>
> Overall, our contribution is a theoretical and geometric analysis of a generalized SupCon objective (allowing arbitrary $g_+$ and $g_-$) that yields a valid MI bound and concrete projection strategies, whereas the prior works focus on specific application domains with empirically learned label prototypes.
>
> ---
>
> ## Q1.
>  We appreciate the request to clarify the connection and to compare more directly with [1] and [2]. Conceptually, as discussed above, both works can be viewed within our projection-based framework as instances where $g_+=g_-$ corresponds to a learned label prototype, and where no explicit adjustment term $R(X,C)$ is included. We will add a dedicated paragraph in the Related Work section that maps their objectives into our notation and discusses how ProjNCE extends them by (i) introducing the MI-based adjustment term and (ii) allowing distinct projections for positive and negative pairs.
>
> Empirically, [1] and [2] are tailored to text and network-intrusion domains with very different data modalities and preprocessing pipelines from our vision/audio benchmarks. Implementing full end-to-end versions of these methods in our settings would therefore require substantial re-engineering. Due to space and computational constraints, we were not able to include such cross-domain reproductions. However, our formulation is generic, and we believe that ProjNCE-style projections and adjustment terms could be fruitfully combined with the label-embedding strategies of [1] and [2].
>
> ---
>
> ## Q2.
>  We are grateful for the question regarding NWJ and stability. Our use of the NWJ bound is primarily analytical. We start from the NWJ variational form of MI because it naturally yields a decomposition in which the supervised InfoNCE term and the adjustment $R(X,C)$ appear in a clean and interpretable way. However, we do not optimize the raw NWJ objective in practice. The actual training loss is ProjNCE objective, with the adjustment term estimated via Nadaraya–Watson smoothing. As a result, ProjNCE inherits the well-known stability properties of InfoNCE, and in our experiments we did not observe the severe instabilities that have been reported for direct NWJ maximization.
>
> The regularization strategies of [3] and [4] are designed to improve the stability and tightness of general MI estimators. These techniques are complementary to our work: in principle, one could apply similar regularization to the underlying MI bound from which ProjNCE is derived. Even without such additional regularization, our practical objective behaves like a standard InfoNCE loss and is empirically stable, as illustrated in the training curves of Figure 2 and Figure 4 (Appendix D.6).
>
> ---
>
> We hope that these clarifications address the reviewer’s concerns and questions, and we would be very happy to provide further details or revisions if anything remains unclear.

---

### Official Review · Reviewer_YBd7 · 2025-10-31

**Soundness:** 3
**Presentation:** 2
**Contribution:** 2
**Rating:** 4
**Confidence:** 4

**Summary:**

This paper proposes ProjNCE, a unified contrastive learning framework that bridges self-supervised (InfoNCE) and supervised contrastive learning (SupCon) through a mutual-information–based formulation. By introducing projection functions and an adjustment term for negative pairs, ProjNCE not only establishes a valid lower bound on mutual information but also enables flexible class embedding strategies such as centroid-based projections. Experiments on image and audio datasets demonstrate that ProjNCE consistently outperforms both SupCon and cross-entropy training, offering a framework that combines theoretical clarity with practical improvements in contrastive learning.

**Strengths:**

- The paper discusses how SupCon relates to mutual information, which is an unexplored and interesting topic.
- The derivation of ProjNCE as a generalized form of InfoNCE is clearly presented and mathematically sound.
- The idea of decoupling projection functions for positive and negative pairs is well motivated.

**Weaknesses:**

- It remains unclear whether the observed improvements stem from the proposed MI-based formulation itself or merely from the additional parameters introduced by the projection functions.
- The practical benefit of maintaining a “valid MI bound” is not convincingly demonstrated—there is no evidence that tighter MI bounds correlate with better downstream performance or robustness.
- The comparisons omit several recent strong baselines trained on larger-scale datasets, which weakens the empirical significance of the reported gains.
- The proposed bound does not empirically demonstrate reduced bias compared to existing methods.

**Questions:**

Can you provide empirical evidence showing that the proposed bound indeed exhibits reduced bias relative to the true mutual information, compared to existing contrastive objectives?

---

> ### Author Response · Authors · 2025-11-21
> **Rebuttal**
>
> We sincerely thank the reviewer for the thoughtful and constructive comments and questions.
> Based on the reviewers’ feedback, we have added several new results: 1) PaCo (SupCon baseline) in Table 3; 2) ablation of $\beta$ (Table 5 in Appendix D.2), embedding dimensionality (Table 6 in Appendix D.3), batch size (Table 7 in Appendix D.4), and temperature (Table 8 in Appendix D.5); and 3) training stability (Figure 4 in Appendix D.6).
>
> Below, we respectfully address the weaknesses (W) and questions (Q) point by point.
>
> ---
>
> ## W1.
> We appreciate this concern. A similar point was also raised by Reviewer 5QFy (Q1). To examine whether the observed gains truly come from the proposed MI-based formulation, we compare SupCon and ProjNCE directly in Table 1. ProjNCE can be viewed as SupCon augmented with the adjustment term $R(X,C)$, and we observe consistent—although sometimes modest—performance improvements. We suspect that SupCon already achieves very strong performance, so the gap between SupCon and the optimal accuracy is small; in such a regime, even a modest but systematic gain is meaningful.
>
> ---
>
> ## W2.
>  A valid MI lower bound is important because it guarantees that minimizing the loss increases a well-defined quantity $I(Z;C)$. When the adjustment $R(X,C)$ is omitted, as in SupCon, this guarantee no longer holds. Empirically, Table 1 provides evidence for the benefit of maintaining a valid bound: incorporating the adjustment term into SupCon (i.e., using ProjNCE) leads to improved accuracy.
>
> From a theoretical perspective, SupCon is not a looser MI bound—it is simply not an MI bound at all in the supervised setting. To the best of our knowledge, ProjNCE is the first objective that provides a valid MI lower bound tailored to supervised contrastive learning, thereby filling this gap.
>
> ---
>
> ## W3.
> We chose our experimental suite to test whether the information-theoretic and projection-based analysis carries over to practice across both vision and audio domains. In particular, we believe that ImageNet (together with CIFAR-10/100 and others) serves as a widely accepted large-scale benchmark that meaningfully validates our analysis, while the audio datasets demonstrate that ProjNCE generalizes beyond images. We certainly agree that additional large-scale settings would be of interest; due to space and computational constraints we could not include more in this submission, but we view such extensions (e.g., larger backbones or additional modalities) as promising directions for future work.
>
> ---
>
> ## W4.
> To the best of our knowledge, ProjNCE is the first supervised contrastive objective that yields a valid MI lower bound. We are not aware of other supervised contrastive methods that explicitly connect to MI in this way. If the reviewer is aware of such methods, we would be very grateful for the pointers and would be happy to investigate and compare them.
>
> Regarding the tightness of the bound, as noted also by Reviewer CpYY, a key ingredient in our derivation is the NWJ bound, which is known to be tighter than several alternative variational MI bounds in the literature. If the reviewer’s concern is about the quality of the MI estimate achieved by ProjNCE in practice, Figure 2 directly addresses this: we perform an MI maximization task comparing InfoNCE, SupCon, and ProjNCE, and observe that ProjNCE attains higher or at least comparable estimated MI on both image and audio datasets. This suggests that the proposed bound provides a better approximation of the true mutual information in these settings.
>
> ---
>
> ## Q1.
> We agree that directly comparing bounds to the true MI is informative. On realistic datasets the true MI is intractable, and hence we estimated MI using MI estimator. Figure 2 provides the MI result from InfoNCE, SupCon, and ProjNCE.
>
> ---
>
> We hope that our responses answer the reviewer’s concern and questions. If any concern remains, please let us know, we would be happy to further discuss.

---

### Official Review · Reviewer_5QFy · 2025-11-01

**Soundness:** 3
**Presentation:** 3
**Contribution:** 3
**Rating:** 4
**Confidence:** 3

**Summary:**

This paper attempts to unify supervised and self-supervised contrastive learning through the ProjNCE loss. The idea behind this unification is to generalize the distance measure in the InfoNCE loss, replacing $⟨f(x_i), c_i⟩$ with $⟨f(x_i), g_{+}(c_i)⟩$ for positive pairs and $⟨f(x_i), g_{-}(c_i)⟩$ for negative pairs, where g₊ and g₋ are projection functions. In classical losses, these functions are typically the identity mapping or the class-average embedding (centroid). The authors also add an additional adjustment term, R(X, C), to establish a valid mutual-information lower bound. Experiments are conducted on vision datasets such as CIFAR, ImageNet, Caltech256, and Food101, as well as on audio datasets including CREMA-D and SpeechCommands. For linear-probe classification, ProjNCE consistently outperforms supervised cross-entropy and SupCon in accuracy across almost all datasets.

**Strengths:**

1. strong linear-probe results on both vision and audio datasets, outperforming CE and SupCon, with notable robustness to noisy labels.

2. the paper evaluates multiple projection strategies (Orthogonal, Median, MLP), showing consistent gains.

3. the derivation of ProjNCE as a proper MI lower bound has a good theoretical motivation.

**Weaknesses:**

1. the paper does not compare against other self-supervised methods such as BYOL, Barlow Twins, DINO, VICReg, or PaCo, which limits its positioning relative to the state of the art.

2. The paper does not include sensitivity studies on key hyperparameters such as $\beta$, the choice of kernel bandwidth, or embedding dimensionality.

3. I am not sure about correctness of the claim on line 154 that SupCon is mathematically equivalent to $I^{\text{self-p}}_{NCE}(Z;C)$ is inaccurate. Substituting $g+(c_i)$ with a class-centroid embedding and $g-(c_j)$ with sample embeddings does not reproduce the pairwise SupCon loss. SupCon averages the negative log conditional probabilities over all *positive pairs*, whereas the projection-based variant contrasts each point with its class centroid. For example, if we have embeddings $a,b$ from class 1 and $c$ from class 2:

SupCon: $\Big[\log\frac{e^{\langle a,b\rangle}}{e^{\langle a,b\rangle}+e^{\langle a,c\rangle}} +\log\frac{e^{\langle b,a\rangle}}{e^{\langle b,a\rangle}+e^{\langle b,c\rangle}}\Big]$ while $I^{\text{self-p}}_{\mathrm{NCE}}= -\Big[\log\frac{e^{\langle a,(a+b)/2\rangle}}{e^{\langle a,b\rangle}+e^{\langle a,c\rangle}}
+\log\frac{e^{\langle b,(a+b)/2\rangle}}{e^{\langle b,a\rangle}+e^{\langle b,c\rangle}}\Big]$

These coincide only when $a = b$ or under a trivial critic and in fact you can bound one to the other with Jensen inequality. The equivalence however holds under alignment and uniformity approximation as the authors motivated their choices with in lines 108-128 (for some common choices of $\psi$).

**Questions:**

1. can you report quantitative results with and without the adjustment term $R(X, C)$ to isolate its contribution?

2. Figure 1 provides qualitative analysis for $\beta$ = 1, 5, 10, but quantitative accuracy trends are missing. How sensitive is performance to $\beta$?

3. What's the motivation for choosing NW over alternatives? How sensitive are results to the kernel bandwidth?

4. in line 410 you state \( K(t) = 1 − t^2 \). What motivated this kernel, and how would other kernels affect convergence or accuracy?

5. how does your proposed objective differ from other frameworks that unified InfoNCE and SupCon such as: UniCL [1], X-Sample Contrastive Loss [2], I-Con [3]?

[1] Yang, Jianwei, et al. "Unified contrastive learning in image-text-label space." CVPR 2022.

[2] Sobal, Vlad, et al. "X-Sample Contrastive Loss: Improving Contrastive Learning with Sample Similarity Graphs." ICLR 2025.

[3] Alshammari, Shaden, et al. "I-Con: A Unifying Framework for Representation Learning." ICLR 2025.

---

> ### Author Response · Authors · 2025-11-21
> **Rebuttal 1/2**
>
> We sincerely thank the reviewer for the thoughtful and constructive comments and questions.
> Based on the reviewers’ feedback, we have added several new results: 1) PaCo (SupCon baseline) in Table 3; 2) ablation of $\beta$ (Table 5 in Appendix D.2), embedding dimensionality (Table 6 in Appendix D.3), batch size (Table 7 in Appendix D.4), and temperature (Table 8 in Appendix D.5); and 3) training stability (Figure 4 in Appendix D.6).
>
> Below, we respectfully address the weaknesses (W) and questions (Q) point by point.
>
> ---
>
> ## W1.
> We are grateful for the comment regarding the positioning of our work.
> The methods mentioned by the reviewer are indeed highly relevant, but we view them as complementary rather than direct competitors to ProjNCE. BYOL, Barlow Twins, DINO, and VICReg are purely self-supervised methods that do not use class labels during pre-training, whereas ProjNCE is specifically designed for the supervised contrastive setting. PaCo is supervised contrastive learning method, which aligns with our paper’s problem. In our empirical study we therefore follow the standard SupCon literature and primarily compare against supervised objectives, now including PaCo [1] (Table 3). ProjNCE still outperforms SupCon and PaCo.
>
> ---
>
> ## W2.
>  We thank the reviewer for highlighting the importance of sensitivity analyses. Our original submission already included a bandwidth ablation in Figure 3 (Appendix D). In the revised version, we now additionally provide ablations over $\beta$ and embedding dimensionality in Tables 5 and 6, respectively. We further include ablations over batch size and the temperature parameter in Tables 7 and 8.
> Consistent with the analysis in Section 2 (Corollary 2.2), we observe that choosing $\beta=1$ yields the best overall performance. Moreover, ProjNCE improves over SupCon for almost all embedding dimensions $d \in \\{16,32,64,128,256 \\}$, indicating that the performance gains are not tied to a particular representation size.
>
> ---
>
> ## W3.
>  We are very grateful to the reviewer for pointing out this subtle but important issue. We carefully re-examined the relationship between SupCon and $I_{\mathrm{NCE}}^{\mathrm{self\text{-}p}}$ with $g_+$ and $g_-$. Our investigation revealed that, at the level of the analytic expressions, SupCon and $I_{\mathrm{NCE}}^{\mathrm{self\text{-}p}}$ can indeed coincide; the discrepancy in our manuscript stems from an imprecise definition of $\mathcal{P}(i)$. In the submitted version, we defined $\mathcal{P}(i)$ as the set of indices corresponding to class $c_i$, whereas in the original SupCon paper $\mathcal{P}(i)$ is defined as the set of all indices with label $c_i$ excluding the index $i$ itself. With a consistent definition of $\mathcal{P}(i)$, SupCon and $I_{\mathrm{NCE}}^{\mathrm{self\text{-}p}}$ become identical under suitable choices of the projections. We have corrected the definition of $\mathcal{P}(i)$ in the revised manuscript.
> Regarding the alignment–uniformity view, we also clarify that the decomposition in Eq. (2) is an exact equality (rather than an approximation). We truly appreciate the reviewer’s careful reading and helpful suggestion here.
>
>
>
> [1] Cui, Jiequan, et al. "Parametric contrastive learning." Proceedings of the IEEE/CVF international conference on computer vision. 2021.

---

> ### Author Response · Authors · 2025-11-21
> **rebuttal 2/2**
>
> ## Q1.
>  We completely agree that isolating the contribution of the adjustment term $R(X,C)$ is important. In response, we have added ablations that compare ProjNCE with and without $R(X,C)$ (i.e., $\beta=0$ vs. $\beta>0$) on CIFAR-10, CIFAR-100, and Tiny-ImageNet; these results are reported in Table 5 (Appendix D.2) and summarized in Table 1 (Section 2.4).
> Empirically, we find that $\beta=1$ generally improves classification accuracy, whereas too large a $\beta$ can over-emphasize the mitigation of false negatives and reduce clean accuracy. This behavior is consistent with Figure 1, where increasing $\beta$ spreads class clusters more evenly but may reduce their compactness.
>
> ---
>
> ## Q2.
> Table 5 provides quantitative results for different choices of $\beta$. The main goal of Figure 1 is to provide a qualitative visualization of how the adjustment term affects the geometry of the embeddings, rather than to optimize accuracy for each setting. In the new ablation table, setting $\beta=1$ achieves the best accuracy, which is in line with our theoretical analysis (see Corollary 2.2). We will make this connection clearer in the revised manuscript.
>
> ---
>
> ## Q3.
>  We appreciate the reviewer’s question regarding the choice of estimator. We employ the Nadaraya–Watson (NW) estimator because it is a classical and widely used kernel estimator in nonparametric regression, with well-understood theoretical guarantees such as consistency. While other estimators could certainly be considered, NW provides a simple and principled choice that integrates naturally into our framework.
> To further address the reviewer’s concern, we also provide an ablation over kernel parameters, including the bandwidth, in Figure 3 (Appendix D). These results indicate that our method is fairly robust to the bandwidth hyperparameter when using common distance metrics such as $\ell_1$ or $\ell_2$.
>
> ---
>
> ## Q4.
> We thank the reviewer for this insightful question. Among several common kernels (uniform, triangular, Gaussian, etc.), we chose the Epanechnikov (parabolic) kernel because it is known to be optimal in a mean-squared-error sense for kernel regression. Of course, other kernels are also viable, and our formulation does not depend on this specific choice. It is generally recognized that the bandwidth is the more critical hyperparameter in kernel methods, whereas the particular kernel shape has a comparatively smaller effect once the bandwidth is chosen appropriately. We will mention this rationale in the appendix for completeness.
>
> ---
>
> ## Q5.
>  We are grateful for the suggestion to discuss UniCL, X-Sample, and I-Con in more detail. These works all aim to unify or generalize contrastive objectives, but along directions that are largely orthogonal to ProjNCE. UniCL unifies different data modalities (image–label and image–text) under a single InfoNCE loss in an image–text–label space, without introducing an adjustment term or analyzing the bias of SupCon as an MI estimator. X-Sample remains within the standard self-supervised InfoNCE framework and modifies the loss by re-weighting sample pairs according to a similarity graph, rather than via an explicit MI decomposition over labels. I-Con provides a broad information-theoretic umbrella that recasts various representation-learning methods as matching neighborhood distributions, but it does not derive a specific connection between supervised InfoNCE and SupCon or introduce an explicit adjustment term.
> In contrast, ProjNCE focuses on the supervised contrastive setting: we show that SupCon implicitly omits the adjustment $R(X,C)$, propose projection functions $g_+$ and $g_-$ that generalize SupCon’s label embeddings, and empirically demonstrate that reinstating a valid MI lower bound leads to improved performance. We have added a dedicated paragraph in the Related Work section to clearly articulate these relationships.
>
> ---
>
> We sincerely hope that our responses address the reviewer’s concerns and questions. In addition to the clarifications above, we have also included an additional baseline, Parametric Contrastive Learning [1], in Table 3. We would be very happy to further revise the manuscript if any points remain unclear.
>
> [1] Cui, Jiequan, et al. "Parametric contrastive learning." Proceedings of the IEEE/CVF international conference on computer vision. 2021.

---

### Author Response · Authors · 2025-12-04
**Final remark**

We thank the Reviewers, Area Chairs, Senior Area Chairs, and Program Chairs for the time, expertise, and constructive feedback.
Our rebuttal and revised manuscript directly target the common themes raised across reviews and their individual comments.

Our revision focuses on the five main themes raised across reviewers.

---

### **1. Contribution and novelty.**

 Several reviewers asked whether ProjNCE is incremental over prior metric-learning or label-representation methods. In the revision we clarified that ProjNCE is derived from a supervised InfoNCE/NWJ mutual information formulation:
* We show that when positive and negative projections differ, an adjustment term $R(X,C)$ necessarily appears in the supervised MI bound. This term is absent in SupCon and in prior label-prototype losses and is not a heuristic reweighting.
* We obtain a generalized SupCon objective with separate projections for positives and negatives, recovering SupCon and prototype-label methods as special cases while allowing distinct label representations for positive and negative pairs.
* We study principled projection choices that target specific properties (e.g., optimal $\ell_2$ projection, robustness, or hyperparameter-free design) instead of learning arbitrary label embeddings.

We added a dedicated Related-Work paragraph explicitly positioning ProjNCE against the cited works and explaining why they are complementary rather than equivalent.

---

### **2. Empirical scope and baselines.**
Concerns about evaluation scale and baselines were also central. Our original submission already included ImageNet-1K and multiple audio datasets, showing generalizability; in the revision we:
* Added PaCo [1] as a strong supervised contrastive baseline (Table 3), where ProjNCE still outperforms SupCon and PaCo.
* Emphasized that our experiments span six image and two audio benchmarks, showing that the gains are not confined to small datasets.
* Clarified that self-supervised methods such as BYOL and VICReg are complementary rather than direct competitors in our supervised setting.

[1] Cui, Jiequan, et al. "Parametric contrastive learning." CVPR. 2021.

---

### **3. What drives the gains? Adjustment term, MI bound, and projections.**
Reviewers asked whether improvements come from the MI-based formulation itself or simply extra flexibility. We therefore added several targeted studies:
* **Ablation of the adjustment term $R(X,C)$** by comparing SupCon ($\beta=0$) and ProjNCE ($\beta=0$) on CIFAR-10/100 and Tiny-ImageNet. We find that $\beta=1$ consistently improves accuracy (aligned with our information-theoretic analysis), while very large $\beta$ can over-spread clusters and hurt performance. This directly addresses the questions around Figure 1 and isolates the effect of the adjustment term.
* **Hyperparameter robustness:** new tables report ablations over $\beta$, embedding dimensionality, batch size, and temperature across multiple datasets. ProjNCE’s advantage over SupCon is robust across these regimes.
* **Estimator and kernel choices:** we explain our use of Nadaraya-Watson estimator and Epanchnikov kernel, and we emphasize the ablation of bandwidth showing that performance is fairly insensitive to bandwidth/kernel variations.

---

### **4. MI bound quality, stability, and relation to NWJ bound.**
Reviewer CpYY raised concerns about the stability of NWJ bound and whether our bound truly helps. In response, we
* clarified that NWJ bound is used analytically to derive ProjNCE; training minimizes the ProjNCE loss, not the raw NWJ objective.
* added training-stability plots (Figure 4) showing no pathological behavior.
* conducted MI-estimation experiments comparing InfoNCE, SupCon, and ProjNCE, observing that ProjNCE attains higher or at least comparable estimated MI. This directly addresses the question of whether the proposed bound better approximates the true mutual information.

---

### **5. Clarifying the SupCon–ProjNCE equivalence claim in eq (5).**
Reviewer 5QFy accurately identified an inaccuracy in our equivalence statement. Upon re-examining the derivation, we confirmed its correctness, but we discovered a discrepancy in the definition of an index set $\mathcal{P}(i)$ between our formulation and the SupCon paper. By adopting the corrected definition, which aligns with the original SupCon paper, we verified that SupCon and our self-projection variant indeed coincide at the analytic level. Consequently, we have updated the definition to reflect the corrected version.

---

We are confident that our response addressed reviewers’ concerns and questions. Despite the Overleaf incident, reviewer 2LFY positively acknowledged our rebuttal and increased their score.

We acknowledge the constructive engagement from all involved in this process and anticipate that the final decision will be informed by the positive developments and strengthened evidence that emerged during the discussion phase.

Best regard,

Authors of submission 16782

---

### Meta-Review · Area_Chair_GXwr · 2026-01-06

**Summary:**

The paper proposes ProjNCE, a projection-based contrastive objective intended to unify self-supervised InfoNCE and supervised contrastive learning (SupCon) under a mutual-information (MI) framework, introducing separate projection functions and an adjustment term to recover a valid MI lower bound in the supervised setting. Reviewers generally found the paper technically sound, clearly written, and empirically well executed, with consistent linear-probe improvements over SupCon and cross-entropy on several vision and audio benchmarks. However, across all reviews there is a common concern that the conceptual and empirical contributions are incremental, with limited evidence that the proposed MI-based formulation offers clear practical or theoretical advantages beyond existing supervised contrastive or prototype-based methods.

**Reviewer Concerns:**

The rebuttal successfully addressed several secondary and technical concerns. In particular, reviewers’ questions regarding missing ablations (e.g., β, embedding dimensionality, batch size, temperature), training stability, kernel choice, and the formal relationship between SupCon and the proposed projection-based formulation were handled carefully and in good faith. The authors also expanded comparisons to PaCo, clarified theoretical definitions, and improved the discussion of related unification frameworks (e.g., UniCL, X-Sample, I-Con), which strengthened the presentation and internal consistency of the work.

However, key concerns remain unresolved. Multiple reviewers questioned whether the observed performance gains stem from the proposed MI-based interpretation itself or simply from additional flexibility introduced by projection functions and weighting terms, a distinction that is not convincingly isolated empirically. The practical benefit of enforcing a “valid MI lower bound” remains largely theoretical; there is no clear evidence that tighter or more principled MI bounds correlate with improved downstream performance, robustness, or generalization beyond modest accuracy gains.

**Reviewer Scores:**

all negative reviews.

---

### Decision · Program_Chairs · 2026-01-26

Reject